# The Antioxidant Potential and Anticancer Activity of *Halodule uninervis* Ethanolic Extract against Triple-Negative Breast Cancer Cells

**DOI:** 10.3390/antiox13060726

**Published:** 2024-06-14

**Authors:** Nadine Wehbe, Adnan Badran, Serine Baydoun, Ali Al-Sawalmih, Marc Maresca, Elias Baydoun, Joelle Edward Mesmar

**Affiliations:** 1Department of Biology, Faculty of Arts and Sciences, American University of Beirut, Riad El Solh, Beirut 1107 2020, Lebanon; nww04@mail.aub.edu (N.W.); eliasbay@aub.edu.lb (E.B.); 2Department of Nutrition, Faculty of Pharmacy and Medical Sciences, University of Petra, Amman 11196, Jordan; abadran@uop.edy.jo; 3Breast Imaging Section, Imaging Institute, Cleveland Clinic Foundation, Cleveland, OH 44195, USA; baydous@ccf.org; 4Marine Science Station, University of Jordan, Aqaba 11942, Jordan; a.sawalmih@ju.edu.jo; 5Aix-Marseille Univ, CNRS, Centrale Marseille, iSM2, 13013 Marseille, France

**Keywords:** *Haludule uninervis*, breast cancer, herbal medicine, antioxidant, malignant phenotype

## Abstract

Natural remedies have been indispensable to traditional medicine practices for generations, offering therapeutic solutions for various ailments. In modern times, these natural products continue to play a pivotal role in the discovery of new drugs, especially for cancer treatment. The marine ecosystem offers a wide range of plants with potential anticancer activities due to their distinct biochemical diversity and adaptation to extreme situations. The seagrass *Halodule uninervis* is rich in diverse bioactive metabolites that bestow the plant with various pharmacological properties. However, its anticancer activity against invasive triple-negative breast cancer (TNBC) is still poorly investigated. In the present study, the phytochemical composition of an ethanolic extract of *H. uninervis* (HUE) was screened, and its antioxidant potential was evaluated. Moreover, the anticancer potential of HUE against MDA-MB-231 cells was investigated along with the possible underlying mechanisms of action. Our results showed that HUE is rich in diverse phytochemicals that are known for their antioxidant and anticancer effects. In MDA-MB-231 cells, HUE targeted the hallmarks of cancer, including cell proliferation, adhesion, migration, invasion, and angiogenesis. The HUE-mediated anti-proliferative and anti-metastatic effects were associated with the downregulation of the proto-oncogenic STAT3 signaling pathway. Taken together, *H. uninervis* could serve as a valuable source for developing novel drugs targeting TNBC.

## 1. Introduction

Nature serves as an inexhaustible reservoir of bioactive metabolites for the development of novel pharmaceuticals. As a source of herbal medicine, plants have become a very feasible alternative for the treatment of various acute and chronic disorders. According to the World Health Organization (WHO), about 80% of the world’s population relies on plants or herbs for medicinal uses [1]. Since ancient civilizations, plant extracts and their derivatives have been used in traditional medicine practices for the treatment of a wide variety of illnesses including arthritis, diabetes, dermatological problems, malaria, tuberculosis, etc. [2]. Plants are considered to be an invaluable source of medicinal compounds owing to the presence of a plethora of secondary metabolites. In addition to their ecological role, these secondary metabolites possess potent therapeutic properties, including antioxidant, antimicrobial, anti-inflammatory, and anticancer properties. The utilization of plants and plant-derived compounds as anticancer agents has been gaining increased attention due to their natural origin, chemical diversity, and low toxicity [3]. In fact, only 29 drugs from 247 anticancer drugs approved in the last 40 years are strictly synthetic [4]. Examples of plant-derived chemotherapeutic agents include the vinca alkaloids, such as vincristine and vinblastine, which are isolated from the Madagascar periwinkle, *Catharanthus roseus*, and used to treat several types of cancer, namely breast cancer, leukemia, and liver cancer [5]. Paclitaxel is also a potent anticancer agent extracted from *Taxus brevifolia* and used against breast, lung, and ovarian cancers [6]. 

Cancer remains one of the most serious challenges to global health and a leading cause of death worldwide, causing an estimated 10 million deaths in 2020 [7]. In particular, breast cancer emerges as the most prevalent, ranking as the fourth leading cause of cancer-related deaths worldwide [7,8]. Over the years, significant progress has been made in the development of various modalities for the treatment of breast cancer. These include surgery, chemotherapy, and radiation therapy [9]. While these treatments aim to eliminate cancer cells, they are often accompanied with a range of side effects, including, but not limited to, nausea, vomiting, inflammation, hormonal imbalances and damage to healthy tissues by being non-selective. Conversely, hormonal and targeted therapies offer directed treatment with fewer side effects. Hormonal therapy targets estrogen and progesterone receptors, while targeted therapy is effective against human epidermal growth factor 2 receptor. Triple-negative breast cancer (TNBC), accounting for approximately 15–20% of all breast cancer cases, lacks these three receptors and remains unresponsive to such therapies [10]. As such, TNBC represents the most aggressive subtype of breast cancer associated with the poorest prognosis [11]. Therefore, an increasing demand persists for alternative treatment strategies, with researchers exploring various avenues, including the utilization of plant-derived therapies. 

The marine environment represents 95% of the biosphere and harbors a wealth of biodiversity, making it a promising candidate in the search of new therapeutic drugs [12]. Of particular interest are seagrasses, the only flowering plants adapted to live submerged in marine waters. Seagrasses play a vital role in the marine ecosystem, serving as habitats and breeding grounds for a myriad of marine organisms, supporting complex food webs, stabilizing sediments, and contributing to water quality [13]. Beyond their ecological prominence, seagrasses have been utilized in traditional folk medicine to treat ailments, such as fever, muscle pain, wounds, and skin infections, among others [14]. *Halodule uninervis*, belonging to the family *Cymodoceaceae*, is an abundant species of seagrass localized in tropical and subtropical coastal regions around the world. Owing to the presence of diverse bioactive metabolites, *Halodule uninervis* extracts have been documented to exhibit various pharmacological properties, including antidiabetic [15,16], antimicrobial [17], antioxidant [18,19], and anticancer properties [18,20]. However, there is limited research available on the potential therapeutic activity of *Halodule uninervis* in combating breast cancer. This prompted the investigation of the anticancer effect of *Halodule uninervis* ethanolic extract against the aggressive TNBC, using human MDA-MB-321 breast cancer cell line as an in vitro model.

In this study, we analyzed the phytochemical composition of *Halodule uninervis* ethanolic extract (HUE) through qualitative assays and analytical examination via HPLC-PDA-MS/MS. We also explored the anticancer potential of HUE, with a particular focus on TNBC using the MDA-MB-231 cell line, by examining its effects on cell proliferation, the induction of apoptosis, and different hallmarks of cancer metastasis including cell migration, adhesion, and aggregation.

## 2. Materials and Methods

### 2.1. Halodule uninervis Ethanolic Extract

*Halodule uninervis* leaves were harvested from the Gulf of Aqaba, Jordan, and identified by Mohammad Al Zein, a plant taxonomist at the Biology Department of the American University of Beirut (AUB). A voucher specimen under the identification number JO 2023-02 has been archived at the Post Herbarium, AUB. The leaves were cleaned, air-dried in the dark at room temperature, and subsequently ground into fine powder. Then, 10 g of the powder was suspended in 80% ethanol for 72 h in the dark. Afterwards, the solution underwent filtration, drying via rotary vacuum evaporation, and lyophilization. The resulting powder (0.6 g) was dissolved in 80% ethanol at a concentration of 100 mg/mL and stored in the dark at 4 °C.

### 2.2. Phytochemical Screening

The chemical composition of the *Halodule uninervis* extract was investigated by performing qualitative tests to detect the presence of primary and secondary metabolites. 

Test for anthocyanins: A total of 0.5 g of HUE was dissolved in 5 mL of ethanol, followed by ultrasonication for 15 min at 30 °C. Subsequently, 1 mL of extract was combined with 1 mL of NaOH (Sigma-Aldrich Co., St. Louis, MO, USA) and heated for 5 min at 100 °C. The presence of anthocyanins was indicated by the appearance of a bluish-green color [21].

Test for anthraquinones: A total of 0.5 g of HUE was dissolved in 4 mL of benzene (Sigma-Aldrich Co., St. Louis, MO, USA). After filtration, 10% ammonia solution was added to the filtrate. The formation of a red or violet color confirmed the presence of anthraquinones [22].

Test for cardiac glycosides: A total of 0.5 g of HUE was dissolved in 5 mL of ethanol, followed by ultrasonication at 30 °C, filtration, and evaporation. The dried extract was then mixed with 1 mL of glacial acetic acid (Sigma-Aldrich Co., St. Louis, MO, USA) and a few drops of 2% FeCl_3_ (Sigma-Aldrich Co., St. Louis, MO, USA). Afterwards, 1 mL of concentrated sulfuric acid (H_2_SO_4_) (Sigma-Aldrich Co., St. Louis, MO, USA) was added to the side of the test tube. The presence of cardiac glycosides was confirmed by the formation of a brown ring [23]. 

Test for essential oils: A total of 0.5 g of HUE was dissolved in 5 mL of ethanol, followed by ultrasonication at 30 °C and filtration. The filtrate was combined with 100 μL of 1 M NaOH. Then, a small amount of 1 M HCl (MERCK, Darmstadt, Germany) was added. The presence of essential oils was established by the formation of a white precipitate [23].

Test for flavonoids: A total of 0.2 g of HUE was mixed with 1 mL of 2% NaOH. A few drops of diluted acid were then added to the mixture once a concentrated, yellow-colored solution was produced. This disappearance of the color in the solution confirmed the presence of flavonoids [24]. 

Test for phenols: A total of 0.5 g of HUE was dissolved in 5 mL of ethanol, followed by ultrasonication at 30 °C and filtration. The filtrate was mixed with 2 mL of distilled water. Afterwards, a small amount of 5% FeCl_3_ was added, resulting in the formation of a dark green color, which implied the presence of phenols [23].

Test for quinones: A total of 0.5 g of HUE was dissolved in 5 mL of ethanol, followed by ultrasonication at 30 °C and filtration. Next, 1 mL of concentrated sulfuric acid (H_2_SO_4_) was added to the filtrate. The presence of quinones was indicated by the formation of a red color [23]. 

Test for resins: A total of 0.5 g of HUE was dissolved in 5 mL of distilled water, followed by ultrasonication at 30 °C and filtration. The turbidity of the filtrate indicated the presence of resins [23]. 

Test for saponins: A total of 0.5 g of HUE was dissolved in 5 mL of distilled water, followed by ultrasonication at 80 °C and filtration. Once the filtrate reached room temperature, it was shaken until a lasting froth formed, implying the presence of saponins [23].

Test for steroids: A total of 0.5 g of HUE was dissolved in 5 mL of ethanol, followed by ultrasonication at 30 °C, filtration, and evaporation. A few milligrams of the obtained powder were combined with 1 mL each of chloroform (Surechem products, Suffolk, UK) and glacial acetic acid. Afterwards, 1 mL of concentrated sulfuric acid (H_2_SO_4_) was added to the side of the test tube. The presence of steroids was indicated by the formation of a green color [23]. 

Test for tannins: A total of 0.5 g of HUE was dissolved in 5 mL of distilled water, followed by ultrasonication at 80 °C and filtration. Five drops of 0.1% FeCl_3_ were added to the filtrate after it had cooled down to room temperature. The presence of tannins was indicated by brownish-green of blue-green coloration [23].

Test for terpenoids: A total of 0.5 g of HUE was dissolved in 5 mL of chloroform, followed by ultrasonication at 30 °C and filtration. Then, 2 mL of concentrated sulfuric acid (H_2_SO_4_) was added to the filtrate. The presence of quinones was confirmed by the formation of a red-brown color [23]. 

### 2.3. Total Phenolic Content (TPC)

The total phenolic content (TPC) of *Halodule uninervis* was assessed using the Folin–Ciocalteu method, as previously described, with minimal modification [25]. HUE was prepared at a stock concentration of 1 mg/mL. Then, 0.5 mL of the extract solution was mixed with 2.5 mL of 0.2 N Folin–Ciocalteu reagent (Sigma-Aldrich Co., St. Louis, MO, USA) and allowed to oxidize for 5 min. Following this, the reaction was neutralized with 2 mL of 75 g/L sodium carbonate (MERCK, Darmstadt, Germany) and then placed in a water bath at 37 °C for 1 h. Absorbance was measured at 760 nm, using gallic acid (Sigma-Aldrich Co., St. Louis, MO, USA) as the standard. TPC was expressed as percentage of total gallic acid equivalents per gram of extract (mg GAE/g). TPC analysis was replicated three times, and the results are presented as mean values ± standard error of the mean (SEM).

### 2.4. Total Flavonoid Content (TFC)

The total flavonoid content (TFC) of *Halodule uninervis* was estimated using the aluminum colorimetric assay, as previously described, with minor adjustments [25]. Next, 0.5 mL of HUE (from a stock of 1 mg/mL) was combined with 0.1 mL of 10% methanolic aluminum chloride (Sigma-Aldrich Co., St. Louis, MO, USA). This mixture was then kept in the dark at room temperature for 30 min. Then, the absorbance was recorded at 415 nm, with quercetin (Sigma-Aldrich Co., St. Louis, MO, USA) serving as a standard. TFC was expressed as mg quercetin equivalents (QE) per gram of extract (mg QE/g). TFC analysis was replicated three times, and data are presented as mean values ± SEM.

### 2.5. HPLC-PDA-MS/MS

The phytochemical analysis of *Halodule uninervis* was conducted using HPLC-PDA-MS/MS. A SHIMADZU LC-MS 8050 (Shimadzu, Japan) LC system equipped with a triple quadruple spectrometer with an ESI source was used. Separation was established using a C18 reversed-phase column (Zorbax Eclipse XDB-C18, rapid resolution, 4.6 × 150 mm, 3.5 µm, Agilent, Santa Clara, CA, USA). Water and acetonitrile (ACN) (0.1% formic acid each) gradients were applied, starting from 5% to 45% ACN over 45 min and then reaching 60% over the last 5 min, with a flow rate of 1 mL/min. An automatic injection of 10 μL of the sample, at a concentration of 10 mg/mL, was performed using the SIL-40C xs autosampler. LC solution 5.109 software (Shimadzu, Kyoto, Japan) was used for instrument control, and the MS operated in negative ion mode. The *m*/*z* range for MS data acquisition was 100 *m*/*z* to 1500 *m*/*z*, using collision energies of 35 eV and 45 eV. Compound identification was achieved by comparing molecular weights, mass fragmentation patterns, and elution orders with those documented in the literature for the plant and genus, as well as with in-house standards [26].

### 2.6. DPPH (α, α-diphenyl-β-picrylhydrazyl) Antioxidant Activity Assay

The antioxidant capacity of HUE was assessed using the free-radical-scavenging activity of DPPH (Sigma-Aldrich Co., St. Louis, MO, USA). Different concentrations of HUE (5, 10, 25, 50, 100, 200, and 400 μg/mL) were added to 0.5 mL of DPPH solution (from a stock of 0.5 mM in methanol) and 3 mL of methanol and incubated in the dark at room temperature for 30 min. The absorbance was then measure at 517 nm using an ELISA microplate reader (Thermo Scientific MULTISKAN GO, Waltham, MA, USA). The DPPH-scavenging activity of each extract concentration was calculated as a percentage inhibition using the following formula: $$\%\;\mathrm{of}\;\mathrm{inhibition}=\frac{absorbance\;of\;blank-absorbance\;of\;extract\;at\;each\;concentration}{absorbance\;of\;blank}\times 100$$
where the blank contained 0.5 mL of 80% ethanol, 0.5 mL of DPPH solution, and 3 mL of methanol. Ascorbic acid was used as a standard. 

### 2.7. Cell Culture

Human breast cancer cells MDA-MB-231 (American Tissue Culture Collection, ATCC, Manassas, VA, USA), human pancreatic cancer cells Capan-2 (Cell Line Service, CLS, Eppenlheim, Germany), and human noncancerous fibroblasts IMR-90 (American Tissue Culture Collection, ATCC, Manassas, VA, USA) were cultured in DMEM high-glucose medium supplemented with 10% fetal bovine serum (FBS) (both from Sigma-Aldrich, St. Louis, MO, USA) and 1% penicillin/streptomycin (Lonza, Visp, Switzerland). Human colorectal cancer cells HCT116 and human prostate cancer cells 22RV1 (both from American Tissue Culture Collection, ATCC, Manassas, VA, USA) were maintained in RPMI-1640 medium supplemented with 10% fetal bovine serum (FBS) (both from Sigma-Aldrich, St. Louis, MO, USA), 1% penicillin/streptomycin (Lonza, Visp, Switzerland), and 1% sodium pyruvate (Sigma-Aldrich, St. Louis, MO, USA). All cells were kept in a humidified incubator at 37 °C and 5% CO_2_.

### 2.8. Cell Proliferation and Cell Toxicity Assays

For proliferation assay, cells were seeded in 96-well plates at a density of 5 × 10^3^ cells per well and incubated for 24 h until they reached 30–40% confluency. Then, cells were treated with increasing concentrations of HUE and incubated for 24, 48, and 72 h. For the cell toxicity assay, cells were seeded in 96-well plates at a density of 20 × 10^3^ cells per well and incubated for 24 h until they reached 100% confluency before being exposed to increasing concentrations of HUE for 72 h. In both cases, at the end of the incubation, cell viability was evaluated using the 3-(4,5-dimethylthiazol-2-yl)-2,5-diphenyltetrazolium bromide (MTT; Sigma-Aldrich, St. Louis, MO, USA) reduction assay. MTT solution (5 mg/mL) was added to the treated cells and incubated for 3 h at 37 °C. Afterwards, DMSO was used to dissolve the formed formazan, and absorbance was measured at 595 nm using an ELISA microplate reader. Cell proliferation and cell viability were expressed as the percentage of the proliferation or viability of the treated cells with respect to the vehicle (ethanol)-treated ones, where the proliferation or viability was assumed to be 100%. The assay was performed in triplicate and repeated three times. Data are presented as mean values ± SEM. In addition to MTT assay, the number of cells was also determined through ATP and protein quantification using CellTiter-Glo (from Promega, Madison, WI, USA) and a BCA assay kit (from Sigma Aldrich, Saint Louis, MO, USA), respectively.

### 2.9. Flow Cytometry Analysis of Cell Cycle

MDA-MB-231 cells were cultured in 100 mm tissue culture plates and subsequently incubated with or without HUE. After 24 h, cells were harvested, washed twice, and suspended in 500 μL PBS. The cells were then fixed with an equal volume of 100% ethanol and kept at −20 °C for a minimum of 12 h. Subsequently, the cells were centrifuged and pelleted, washed twice with PBS, resuspended in PBS containing 4′, 6-diamidino-2-phenylindole, dihydrochloride at 1 μg/mL (DAPI, Cell signaling Technology, Inc., Danvers, MA, USA), and kept at room temperature for 30 min. The cell samples were then assessed using the BD FACSCanto II Flow Cytometry System (Becton Dickinson, Franklin Lakes, NJ, USA), with data acquisition facilitated by FACSDiva 6.1 software.

### 2.10. Microscopic Analysis of Apoptotic Morphological Changes

MDA-MB-231 cells were seeded in 6-well plates at a density of 2 × 10^5^ and then incubated in the presence or absence of the indicated concentrations of HUE. Following a 24 h incubation time, an inverted phase-contrast microscope was used to detect morphological changes indicating apoptotic cells. Images were taken at 10×, 20×, and 40× magnifications. 

4′, 6-diamidino-2-phenylindole, dihydrochloride (DAPI) staining was used to visualize changes in nuclear morphology. For this, 8 × 10^4^ cells were seeded in 12-well plates with or without HUE for 24 h. Then, 4% formaldehyde was used to fix the cells, followed by DAPI staining (1 μg/mL). Cells were then visualized using fluorescence microscopy. 

### 2.11. Wound-Healing Assay

MDA-MB-231 cells were seeded in 12-well plates at a density of 4 × 10^5^ and incubated for 24 h until they reached confluency. The confluent monolayer was then scratched by a 10 μL pipette tip. Cells were washed twice with PBS to remove cellular debris before adding fresh medium with or without the indicated concentrations of HUE. Micrographs were taken at the time 0 h (baseline) and at different time points post-scratch (6, 9, and 12 h) using an inverted phase-contrast microscope at 4× magnification. The width of the wound was measured and expressed as the average difference ± SEM between the measurements taken at time zero and the corresponding time points. The assay was repeated three times, and data are presented as mean values ± SEM. 

### 2.12. Trans-Well Migration Chamber Assay

Trans-well inserts (8 μm pore size; BD Biosciences, Bedford, MA, USA) were used to evaluate the migratory abilities of MDA-MB-231 cells. Indeed, 3 × 10^5^ cells were seeded into the upper chamber of the insert. Cells were then treated with or without the indicated concentrations of HUE. DMEM medium supplemented with 10% FBS was used as a chemoattractant in the lower chamber. Cells were then incubated at 37 °C and allowed to migrate for 24 h. Cells that stayed on the upper surface of the insert were removed with a sterile cotton swab. And cells that migrated to the lower surface of the insert were fixed with 4% formaldehyde, stained with DAPI (1 μg/mL), and viewed under a fluorescence microscope at 10× magnification for quantification. The assay was repeated three times, and data are presented as mean values ± SEM.

### 2.13. Matrigel Invasion Assay

A BD Matrigel Invasion Chamber (8μm pore size; BD Biosciences, Bedford, MA, USA) was used to assess the invasive abilities of MDA-MB-231 cells. The experiment is similar to the trans-well migration chamber assay, with the exception of an added Matrigel matrix (diluted 1:20). Cells that invaded the Matrigel layer to the lower surface of the insert were fixed with 4% formaldehyde, stained with DAPI, and viewed under a fluorescence microscope for quantification. The assay was repeated three times, and data are presented as mean values ± SEM.

### 2.14. Adhesion Assay

MD-MB-231 cells were cultured either with the vehicle-containing control or the indicated concentrations of HUE for 24 h. The cells were then seeded at a density of 5 × 10^4^ onto a 24-well plate pre-coated with collagen and allowed to adhere for 3 h. Afterwards, the cells were washed with PBS to remove unattached cells, and the number of attached cells was determined using the MTT reduction assay (described above). 

### 2.15. Aggregation Assay

MDA-MB-231 cells were collected from confluent plates using 2 mM EDTA in Ca^2+^/Mg^2+^-free PBS. Cells were then seeded on 60 mm plates at a density of 4 × 10^5^ with or without the indicated concentrations of HUE. Cells were incubated at 37 °C on a shaker (90 rpm) for 4 h and then fixed with 1% formaldehyde. Images from a minimum of 5 different fields were taken at 4× magnification, and percentage of aggregation was calculated using the following formula: $$\%\;\mathrm{of}\;\mathrm{aggregation}=1-\frac{Nt}{Nc}\times 100$$
where *Nt* is the number of single cells in the treated group, and *Nc* is the number of single cells in the control group. 

### 2.16. Chorioallantoic Membrane Assay

Fertilized chicken eggs were cleaned with 70% ethanol and incubated with rotation at 37 °C with 50% relative humidity. After one week, the highly vascularized chorioallantoic membrane (CAM) was dropped by creating an opening in the eggshell, exposing the air sac. To examine the effect of HUE on blood vessel growth, different concentrations were added onto the CAM and incubated for 24 h. Afterwards, images of the CAM were captured and evaluated using AngioTool 0.5a software to measure the vessels’ lengths and count the number of junctions. 

### 2.17. Whole-Cell Extracts and Western Blotting

To collect whole-cell lysates, MDA-MB-231 cells were washed with PBS and lysed using a lysis buffer (pH 6.7) containing 2% SDS and 60 mM Tris. Afterwards, the lysate was centrifuged at 15,000× *g* for 10 min. The concentration of proteins in the supernatants was determined using the Bradford protein assay kit (Bio-Rad, Hercules, CA, USA). A total of 25–30 μg of the protein extracts were separated by 10% sodium dodecyl sulfate-polyacrylamide gel electrophoresis and then transferred to a polyvinylidene difluoride membrane (Immobilon PVDF; Bio-Rad, Hercules, CA, USA). Next, 5% nonfat dry milk in TBST (TBS and 0.05% Tween) was subsequently used to block the membrane for 1 h at room temperature. Immunodetection involved incubating the membrane with the specified primary antibodies, followed by the corresponding secondary antibody. Immunoreactive bands were visualized using the Clarity Western ECL substrate kit (Bio-Rad, Hercules, CA, USA), as per the manufacturer’s instructions. All primary and secondary antibodies were obtained from Cell Signaling (Cell Signaling Technology, Inc., Danvers, MA, USA). Experiments were replicated three times, and data are presented as mean values ± SEM. 

### 2.18. Statistical Analysis

Statistical analysis of data was carried out using Student’s t-test, a one-way ANOVA (with Dunnett’s post hoc test), or a two-way ANOVA (with the Tukey–Kramer post hoc test). Data are presented as mean ± SEM. A *p*-value of less than 0.05 was considered as statistically significant.

## 3. Results

### 3.1. Analysis of HUE by HPLC-PDA-MS/MS

The bioactive metabolites present in the ethanolic extract obtained from *Halodule uninervis* were tentatively determined via high-performance liquid chromatography coupled with a photodiode array and mass spectrometry detectors (HPLC-PDA-MS/MS). Eighteen compounds were detected in total (Table 1 and Figure 1 and Appendix A). 

### 3.2. Phytochemical Screening

Qualitative phytochemical screening of *H. uninervis* crude extract indicated the presence of several primary and secondary bioactive metabolites, including anthraquinones, cardiac glycosides, flavonoids, phenols, resins, steroids, tannins, and terpenoids (Table 2). 

TPC and TFC assays revealed that HUE is rich in polyphenolic compounds and flavonoids. TPC and TFC were measured to be 321.25 ± 5.86 mg GAE/g extract and 80.05 ± 4.12 mg QE/g extract, respectively.

### 3.3. HUE Has High Antioxidant Capacity

The antioxidant capacity of the HUE crude extract was assessed using the DPPH free-radical-scavenging assay. Our results show significant antioxidant activity and scavenging effects on the DPPH radical with a half-maximum inhibitory concentration (IC_50_) of 301.31 μg/mL compared to an IC_50_ of 8.98 μg/mL obtained with the reference compound ascorbic acid (Figure 2).

These results are concentration-dependent, where the DPPH free radical scavenging of HUE at concentrations of 25, 50, 100, 200, and 400 μg/mL was 0.36 ± 1.87, 6.08 ± 1.7, 17.44 ± 5.3, 34.36 ± 3.2%, and 59.28 ± 1.16, respectively. 

### 3.4. HUE Inhibits the Proliferation of Cancer Cells

HUE was screened for its anti-proliferative activities against different human cancer cells lines: MDA-MB-231, Capan-2, HCT116, and 22RV1. The effects of various concentrations of HUE (0, 50, 100, 200, 400, and 600 μg/mL) were assessed at 24, 48, and 72 h of treatment. The results revealed that HUE decreased cell proliferation in a concentration- and time-dependent manner in all studied cell lines (Appendix A and Appendix A).

For the purpose of this study, we focused on the MDA-MB-2321 TNBC cell line. After 48 h of treatment with 50, 100, 200, 400 and 600 μg/mL of HUE, the proliferation of MDA-MB-231 cells was 71.03 ± 4.08, 65.9 ± 4.25, 52.72 ± 6.09, 25.80 ± 5.0, and 20.50 ± 3.72% of that of the control cells, respectively (Figure 3A).

Using the MTT assay, the IC_50_ was 525.34, 173.37, and 146.74 µg/mL at 24, 48, and 72 h. Based on the IC_50_ values, 100 and 200 μg/mL HUE were used in further experiments. In order to confirm that the observed decrease in MTT metabolization corresponds to a real decrease in cell number and not to an enzymatic inhibition of the cellular dehydrogenases by HUE, other assays were performed. The quantification of ATP (by CellTiter-Glo) and protein (by BCA assay) contents in MDA-MB-231 cells exposed for 72 h to HUE at 200 µg/mL showed 59.92 +/− 5.62 and 52.63 +/− 7.98% decreases compared to a 58.22 +/− 6.61% decrease for the MTT signal, demonstrating that HUE indeed inhibits cell division rather than inhibiting cellular dehydrogenases.

To further confirm the anti-proliferative activity of HUE, protein lysates from HUE-treated MDA-MB-231 cells were immunoblotted with an antibody against Ki67, which is highly associated with cancer cell proliferation. Ki67 is a known predictive and prognostic marker for many cancers, and it is highly expressed in TNBC (22). Figure 3B indicates that the treatment of MDA-MB-231 cells with HUE at 100 and 200 μg/mL significantly decreased Ki67 protein levels in a concentration-dependent manner by 0.78- and 0.68-fold compared with the vehicle-treated control cells, respectively. These data confirm that HUE impedes the growth of MDA-MB-231 cells by interfering with their cell proliferation process.

The selectivity of HUE was evaluated by testing its effect on normal human fibroblast cells (IMR-90) and comparing it to the commonly used anti-proliferative drug doxorubicin (DOXO). Our results showed that HUE displays some selectivity against MDA-MB-231 cells over IMR-90 cells, with IC_50_ values of 157.02 μg/mL and 290.05 μg/mL, respectively (Figure 3C). It is worth mentioning that DOXO did not display any selective anti-proliferative effect, with IC_50_ values of 2.5 ng/mL and 1.3 ng/mL for MDA-MB-231 and IMR-90 cells, respectively (Figure 3C).

The cytotoxic effect of HUE was further investigated. Figure 3D shows that HUE has a strong and selective cytotoxic effect on cancer cells, with a half-maximum cytotoxic concentration (CC_50_) equal to 253.8 μg/mL on MDA-MB-231 cells compared to 481.79 μg/mL on IMR-90 cells, demonstrating again the selectivity of HUE over cancer cells. The higher values of CC_50_ compared to IC_50_ for proliferation confirmed that HUE, as with DOXO, is more anti-proliferative than cytotoxic, with the inhibition of the cell division occurring at doses not causing cell death. Importantly, as observed in the anti-proliferation assay, DOXO was found to be more toxic on human normal fibroblasts, with a CC_50_ of 54.3 ng/mL on IMR-90 cells compared to 328 ng/mL on MDA-MB-231 cells (Figure 3D).

### 3.5. HUE Induces Cell Cycle Arrest of MDA-MB-231 Cells in the G_0_/G_1_ Phase

To investigate the mechanism of the HUE-induced anti-proliferative effect on MDA-MB-231 cells, the cell cycle distribution was examined via flow cytometry following 24 h of treatment with 100 μg/mL of HUE. The percentage of cells in G_0_ phase increased in HUE-treated cells (9.9 ± 0.08 vs 24.1 ± 3.06 in control cells), as shown in Figure 4A. This was accompanied with a decrease in the percentage of cells in the G_1_ phase (47.4 ± 2.72 vs 37.2 ± 2.2 in control cells). The results suggest that HUE induces an arrest at the G_0_/G_1_ phase of the cell cycle. Furthermore, the increase in the percentage of cells in the G_0_ phase could indicate the induction of apoptosis (Figure 4A).

The p38 mitogen-activated protein kinase (MAPK) pathway plays a significant role in the regulation of cell cycle progression and induction of apoptosis [27]. Western blotting analysis of the protein levels of the active phosphorylated form of p38 revealed the concentration-dependent activation of p38 in HUE-treated MDA-MB-231 cells, achieving a significant increase at 200 μg/mL (1.51 ± 0.09-fold change in the control) (Figure 4B). In addition, HUE treatment led to a significant increase in the protein levels of the cell cycle regulator p21, a downstream effector of p38 [28,29] (Figure 4B). These results further confirm the cell cycle data. 

P53 is a well-known tumor suppressor that regulates cell cycle by activating p21 [30]. To further investigate the molecular signaling behind HUE-induced cell cycle arrest, the protein levels of phosphorylated p53 were assessed. HUE treatment increased the levels of phosphorylated p53 in a concentration-dependent manner, achieving a significant increase at 200 μg/mL (1.31 ± 0.07-fold change in the control) (Figure 4B). Altogether, these data suggest that the activation of p38 and p53 by HUE could induce cell cycle arrest by regulating p21 activity.

### 3.6. HUE Induces Intrinsic Apopotosis in MDA-MB-231 Cells

MDA-MB-231 cells were treated with HUE for 24 h, then examined using an inverted phase-contrast microscope. The morphological observation of HUE-treated cells showed a concentration-dependent reduction in the total number of cells per microscopic field. In addition, HUE-treated cells showed features that are characteristic of apoptosis (Figure 5A). 

Higher magnifications revealed cells with cytoplasmic shrinkage, loss of epithelial morphology, membrane blebbing, and rounded shape, as well as the appearance of apoptotic bodies (Figure 5A). Further analysis of HUE-treated cells with DAPI staining revealed nuclear condensation, chromatin lysis, and the appearance of apoptotic bodies (Figure 5B).

To unravel the mechanism of apoptosis in HUE-treated MDA-MB-231 cells, we first examined the protein levels of procaspase-3, which is cleaved into the active caspase-3 upon the activation of the intrinsic apoptosis pathway. Our findings indicated that the protein levels of procaspase-3 were decreased significantly in cells treated with 100 and 200 μg/mL HUE (0.86 ± 0.08- and 0.52 ± 0.03-fold reductions, respectively). This suggests that HUE triggers the proteolytic cleavage of procaspase-3, subsequent caspase activation, and the induction of the intrinsic apoptosis pathway (Figure 5C). 

B-cell lymphoma 2 (Bcl-2), an anti-apoptotic protein, is also known to play a crucial role in the intrinsic apoptosis pathway. Our results revealed that HUE decreased the protein levels of Bcl-2 in a concentration-dependent manner, with 200 μg/mL of HUE having a significant difference from the control (Figure 5C). Another regulator of apoptosis is Bcl-2-associated X-protein (Bax), a pro-apoptotic protein. In our study, levels of Bax were significantly upregulated in cells treated with 200 μg/mL HUE (Figure 5C). Altogether, these data strongly indicate that HUE mediates its anticancer activities by targeting apoptotic mechanisms.

### 3.7. HUE Inhibits Migration and Invasive Properties of MDA-MB-231 Cells

Cell migration is a crucial mechanism in many physiological processes such as immune responses, tissue formation, and wound healing. However, aberrant cell migration contributes to cancer metastasis, where tumor cells disseminate from the primary tumor site and colonize in other locations. Here, the effect of HUE on the migration of MDA-MB-231 cells was assessed using the wound-healing and trans-well migration assays. Figure 6A indicates that HUE decreased the ability of cells to migrate and cover the scratched area in a concentration- and time-dependent manner. 

For instance, 12 h after the cell monolayer was scratched, the migration of MDA-MB-231 cells treated with 100 or 200 μg/mL HUE was 0.67 ± 0.09- and 0.30 ± 0.01-fold that of vehicle-treated cells, respectively. The trans-well migration assay further validated these findings. HUE treatment significantly decreased the cell migration ability of MDA-MB-231 cells crossing from the upper to the lower chamber (Figure 6B). 

The invasion of other tissues is a crucial step in the early stages of the cancer metastatic cascade, where cancer cells spread from the primary tumor site and invade secondary sites. Using the Matrigel-coated trans-well chambers, HUE treatment significantly reduced the invasive potential of MDA-MB-231 cells in a dose-dependent manner by 0.55- and 0.18-fold for 100 μg/mL and 200 μg/mL, respectively, compared to the control (Figure 7).

### 3.8. HUE Decreases Adhesion of MDA-MB-231 Cells to Collagen

The ability of tumor cells to adhere to the extracellular matrix (ECM) is another important step in cancer progression and metastasis. To this end, we assessed the effect of HUE on the adhesive ability of MDA-MB-231 cells to collagen, an ECM protein. Figure 8A shows that the treatment of cells with 100 and 200 μg/mL HUE significantly diminished their adhesive abilities to 57.49 ± 4.73 and 29.35 ± 1.81% those of control cells, respectively.

The interaction of cells with their ECM is mediated by a family of adhesion molecules, mainly integrins. High levels of integrin β1 ensure the survival of cancer cells and are associated with their migratory and metastatic abilities [31]. Here, we report that the treatment of MDA-MB-231 cells with 100 and 200 μg/mL HUE significantly decreased integrin β1 protein levels in a concentration-dependent manner by 0.62- and 0.46-fold compared with the vehicle-treated control cells, respectively (Figure 8B). Together, these results confirm that HUE impairs the integrin–ECM axis, which could inhibit the metastatic potential of MDA-MB-231 cells.

### 3.9. HUE Increases Aggregation of MDA-MB-232 Cells

One of the hallmarks of cancer progression towards metastasis is epithelial–mesenchymal transition (EMT), a process wherein epithelial cells acquire a mesenchymal phenotype characterized by a decrease in cell–cell adhesion and enhanced migratory and invasive capabilities. To evaluate the effect of HUE on the cell–cell adhesion properties of MDA-MB-231 cells, an aggregation assay was performed. Figure 9A shows that the treatment of cells with 100 and 200 μg/mL HUE resulted in a concentration-dependent increase in cell–cell aggregates, with significant 39.07 ± 1.71 and 51.87 ± 2.03% increases, respectively.

Occludin is a transmembrane protein localized at tight junctions that plays an important role in cell–cell adhesion. The dysregulation of occludin is associated with increased tumor progression and metastasis [32]. Here, the treatment of MDA-MB-231 cells with 100 and 200 μg/mL HUE increased occludin protein levels in a dose-dependent manner, with a noticeable difference from the control at 200 μg/mL, as observed based on a 1.36 ± 0.10-fold change (Figure 9B).

### 3.10. HUE Inhibits Angiogenesis In Ovo and Decreases Levels of iNOS and COX-2

The dissemination of tumor cells from the primary tumor site and their metastasis into secondary sites entail the formation of new bloods vessels through angiogenesis to supply cells with oxygen and nutrients. Therefore, inhibiting angiogenesis is essential to limit metastasis. To examine HUE affects angiogenesis, the chick-embryo chorioallantoic-membrane (CAM) assay was conducted by applying HUE to the surface of the extensively vascularized CAM membrane for a duration of 24 h. As shown in Figure 10A, HUE treatment significantly inhibited angiogenesis by decreasing the total vessel length and total number of junctions. 

For instance, 100 μg/m HUE caused a reduction in the total vessels length by 27.68 ± 7.18%, as well as a decrease of 11.74 ± 9.09%, compared to the control. 

The anti-angiogenic effects of HUE were further explored by examining its effect on the production of inducible nitric oxide synthase (iNOS), a key producer of nitric oxide (NO), and cyclooxygenase 2 (COX-2), an enzyme that produces prostaglandin E2 (PGE2). Both NO and PGE2 serve as well-known mediators of angiogenesis [33,34]. Our results show that 100 or 200 μg/mL HUE significantly reduced the protein levels of iNOS by 0.72 ± 0.03- and 0.58 ± 0.06-fold, respectively, compared to the vehicle-treated control (Figure 10B). Moreover, the protein levels of COX-2 were decreased by 0.82 ± 0.43- and 0.61 ± 0.03-fold compared to the control after treatment with 100 or 200 μg/mL of HUE, respectively (Figure 10B). These results indicate that HUE inhibits angiogenesis by targeting the production of NO and PGE2.

### 3.11. HUE Inhibits the STAT3 Pathway

STAT3 is a well-recognized key player in tumorigenesis because of its role in promoting cell proliferation, inhibiting apoptosis, and stimulating the migration and invasion of cancer cells [35]. Therefore, targeting the STAT3 pathway is a promising strategy for the development of novel cancer drugs. To investigate whether *H. uninervis* mediates its anticancer activity via targeting the STAT3 pathway, we evaluated the level of the active phosphorylated form of STAT3 in HUE-treated MDA-MB-231 cells. Our results showed that HUE treatment at 100 and 200 μg/mL markedly reduced the levels of STAT3 by 0.83 ± 0.09- and 0.72 ± 0.06-fold, respectively, relative to the control (Figure 11). 

This result suggests that the HUE-mediated effect of TNBC progression and metastasis involves the downregulation of the STAT3 signaling pathway. 

## 4. Discussion

Currently, there is growing interest in screening plants to identify therapeutic agents. However, there are scarce investigations on the therapeutic potential of seagrasses such as *Halodule uninervis*, namely their antioxidant and anticancer activities. In the present study, we evaluated the antioxidant potential of the ethanolic extract of *H. uninervis* (HUE) and analyzed its phytochemical composition qualitatively and quantitatively via HPLC-PDA-MS/MS. Moreover, we investigated the anticancer potential of HUE by assessing its effect on major hallmarks of cancer, including proliferation, migration, invasion, and angiogenesis, against the aggressive TNBC subtype.

The qualitative phytochemical analysis of HUE indicated the presence of several classes of phytochemical compounds, including cardiac glycosides, essential oils, flavonoids, phenols, tannins, and terpenoids (Table 2). Our results are in agreement with previous studies reporting the presence of these phytochemical compounds in *H. uninervis* extracts [18,20] and in *H. pinifolia*, another species of *Halodule* [36,37,38]. Polyphenols, including flavonoids, are well known for their antioxidant, anti-inflammatory, cardioprotective, and neuroprotective activities. As such, the antioxidant potential of *H. uninervis* has previously been documented [18,19]. In accordance, the ethanolic extract of *H. uninervis* exhibited high antioxidant potential in vitro by its ability to scavenge DPPH radicals. Investigating which key proteins are involved in mediating the observed antioxidant potential of HUE is required to understand the underlying molecular mechanism of action. This remains to be tested in future studies. In addition to their antioxidant role, phenols and flavonoids are also known for their anticancer activity [39,40,41]. 

The HPLC-PDA-MS/MS analysis of HUE revealed the presence of 18 different bioactive metabolites (Table 1, Figure 1 and Appendix A), several of which are phenols and flavonoids that have been associated with pharmacological activities. For instance, apigenin, a major phytoconstituent in HUE, is a natural flavonoid commonly found in plants, fruits, and vegetables [42]. Apigenin is known for its diverse pharmacological properties and has previously been reported to exhibit antioxidant [43,44], anti-inflammatory [45,46], and neuroprotective activities [47]. Moreover, apigenin has been shown to have anticancer activities against conditions including lung cancer [48,49], melanoma [50,51], and prostate cancer [52,53], among others. With regard to breast cancer, its anticancer activity has been documented against hormone-positive, as well as triple-negative, breast cancer subtypes [54,55,56]. Other compounds present in HUE with previously reported anti-breast cancer activity include acacetin [57], which is a derivative of apigenin; coumaric acid [58]; kaempferol [59]; and vanillic acid [60]. Thus, given the presence of these bioactive metabolites in HUE, it would be noteworthy to investigate its anticancer potential.

HUE significantly inhibited the cell proliferation of MDA-MB-231 in a dose-dependent manner (Figure 3A). Previous studies have reported the anti-proliferative effects of *H. uninervis* extracts on different cancer cell lines [18,20]. It is worth noting that the anti-proliferative effects of different seagrasses, including *Syringodium isoetifolium*, *Cymodocea serrulata*, *Halophila ovalis*, and *Halodule pinifolia* (a different species of *H. uninervis*), on the ER-positive breast cancer subtype are well documented [61,62,63,64]. However, this is the first study to assess the effects of *H. uninervis* ethanolic extract (HUE) on the triple-negative subtype of breast cancer. The proliferation marker Ki67 is prominently expressed in TNBC and is correlated with its aggressive pathological characteristics [65]. HUE decreased the levels of Ki67 (Figure 3B), supporting the potential use of *H. uninervis* as a source for developing therapeutic agents against TNBC. HUE also induced cell cycle arrest in MDA-MB-231 cells specifically at the G1 phase (Figure 4A). Importantly, our results showed that HUE also exhibits a selective activity on MDA-MB-231 cells compared to the normal fibroblast cells IMR-90. 

Moreover, to gain insight into the mechanism of HUE-induced anti-proliferative effects, the involvement of the p38 mitogen-activated protein kinase (MAPK) pathway was investigated. P38 is known to play a crucial role in maintaining cellular homeostasis by regulating various cellular processes, including cell cycle progression, apoptosis, and cellular response to stress [27]. Our results showed that HUE increased the levels of p38 (Figure 4B), which is in accordance with other studies reporting the relationship between p38 activation and the induction of cell cycle arrest [66,67,68]. In addition, the levels of the cell cycle inhibitor protein p21, a downstream effector of p38 [29], were significantly increased by HUE (Figure 4B). This further verifies the role of the p38 signaling pathway in HUE-induced cell cycle arrest.

P38 could also activate p53, a tumor suppressor protein. P53, referred to as the “guardian of the genome”, plays a critical role in regulating cell cycle progression, DNA repair, and apoptosis in response to cellular stress [69]. Mutations in p53 gene are associated with many cancers and occur in approximately 80% of TNBC cases, contributing to tumor aggressiveness and resistance to therapy [70]. As such, targeting the mutated p53 and restoring its wild-type form is a promising strategy in the development of TNBC-targeted therapeutics. The phosphorylation of p53 at ser15 is a critical post-translational modification that regulates p53 stability and activity [71]. Indeed, studies have shown that phosphorylation of mutant p53 at ser15 restores its tumor-suppressor activity in TNBC [25,72,73]. In accordance, HUE induced the phosphorylation of p53 in MDA-MB-231 cells (Figure 4B), possibly restoring its conformation to the wild-type form and increasing its transcriptional activity. Co-targeting p38 and p53 pathways in TNBC could, therefore, represent a promising therapeutic strategy in halting cancer cell proliferation.

In addition to blocking uncontrolled proliferation, triggering apoptosis in cancer cells is a crucial strategy in cancer therapy [74]. Cancer cells, including TNBC, often evade apoptosis, leading to tumor progression and treatment failure [75,76]. Therefore, inducing apoptotic pathways proves to be a potent approach in cancer treatment. In our study, HUE induced intrinsic apoptosis, as indicated by the activation of caspase-3, the downregulation of the anti-apoptotic Bcl-2 protein, and the upregulation of the pro-apoptotic Bax protein (Figure 5). This is in line with other studies reporting the pro-apoptotic effects of different seagrasses, *Syringodium isoetifolium* and *Halophila ovalis*, on the MCF-7 breast cancer cell line [61,63]. This is also similar to results obtained with other plant extracts, including *Ziziphus nummularia* and *Origanum syriacum*, on TNBC [25,77]. It is worth mentioning that HUE did not induce the production of reactive oxygen species (ROS) (Appendix A), suggesting that *H. uninervis* does not exert cellular oxidative stress.

This is particularly intriguing as excessive ROS production by conventional anticancer drugs is associated with the induction of cell death through apoptosis. However, several anticancer drugs have been described as exerting their effect through the induction of ROS-independent apoptosis in cancer cells while having minimal side-effects on normal cells [78]. Overall, our results indicate a potential anti-proliferative role of *H. uninervis* against TNBC possibly by inducing p38-mediated cell cycle arrest and promoting intrinsic apoptotic pathway in a ROS-independent mechanism. 

Cancer metastasis is a complex process involving the spread of cancer cells from the primary tumor site to secondary organs. It is involved in cancer progression and is responsible for a substantial proportion of cancer-related deaths. TNBC metastasis represents a significant clinical challenge due to its aggressive behavior, poor prognosis, and treatment resistance [79]. Cell migration and invasion are key components of cancer metastasis. Targeting their underlying mechanisms is essential for inhibiting metastatic spread and overcoming resistance to treatment. The dysregulation of cell–cell adhesion and degradation of the extracellular matrix (ECM) by metalloproteinases (MMPs) contribute to the augmentation of cancer cell migration and invasion [80,81]. In our study, the effect of HUE on the migration and invasion properties of MDA-MB-231 was investigated using wound-healing and Boyden chamber assays. HUE significantly inhibited TNBC cell migration and invasion (Figure 6 and Figure 7). MMP-9 and MMP-2 are overexpressed in breast cancer, leading to a higher incidence of metastasis [82]. Several studies of different plant extracts showed that the reduction in MMP-2 and MMP-9 expression inhibits the migratory and invasive capabilities in MDA-MB-231 cells [25,77,83,84]. Whether HUE-induced inhibition of migration and invasion is mediated by the downregulation of MMPs remains to be tested in further studies. 

Cancer metastasis also depends on the interaction of cancer cells with the extracellular matrix (ECM), which mediates various stages of the metastatic cascade [85]. Cancer cells detach from the primary tumor site and invade through the surrounding ECM by interacting with different components of the ECM, including fibronectin, laminin, and collagen [86]. Here, HUE treatment decreased the capacity of MDA-MB-231 cells to adhere to collagen, indicating the disruption of cell–ECM interaction (Figure 8A). Moreover, HUE decreased the levels of integrin β1 (Figure 8B), an adhesion molecule associated with the increased aggressiveness and invasiveness of TNBC [87]. These results confirm the ability of HUE to inhibit the metastatic potential of TNBC cells by targeting cell–cell and cell–ECM interactions.

Epithelial–mesenchymal transition (EMT) has been implicated as a critical mechanism driving TNBC tumor metastasis [88,89]. EMT is characterized by a decrease in cell–cell interactions and increases in cell migration and invasion [90,91]. The dysregulation of key molecular players including E-cadherin, a cell adhesion molecule, and occludin, a protein involved in the formation of tight junctions, has been linked to increased tumor aggressiveness in breast cancer patients [32,92]. In our study, HUE significantly enhanced the cell–cell adhesion of MDA-MB-231 cells in a concentration-dependent manner (Figure 9A). This coincided with an increase in the protein levels of occludin, indicating that HUE could inhibit tumor migration by inducing cell–cell adhesion (Figure 9B). Overall, HUE may potently attenuate TNBC metastasis by impairing cell–ECM interaction, inhibiting cell migration and invasion, and enhancing cell–cell homotypic adhesion.

Angiogenesis, the process of forming new blood vessels, plays a crucial role in tumor growth, invasion, and metastasis [93,94]. In particular, TNBC tumors are often highly angiogenic, facilitating tumor growth and supporting metastatic dissemination [95]. Therefore, impeding angiogenesis by targeting pro-angiogenic factors has emerged as a promising therapeutic strategy to inhibit TNBC tumor growth and metastasis. Both inducible nitric oxide synthase (iNOS) and cyclooxygenase-2 (COX-2) have been implicated in inducing angiogenesis in TNBC [96,97,98,99]. iNOS and COX-2 are enzymes involved in the production of nitric oxide (NO) and prostaglandin E2 (PGE2), respectively, in response to external stimuli present in the tumor microenvironment. NO and PGE2 are potent mediators of angiogenesis, and their upregulation in TNBC promotes metastasis [100,101]. Indeed, HUE treatment was associated with decreases in the levels of iNOS and COX-2 in MDA-MB-231 cells (Figure 10B). Additionally, HUE inhibited angiogenesis successfully by decreasing vessel length and number of junctions on the chick-embryo CAM (Figure 10A). In accordance, the seagrass *Thalassia testudinum* was found to suppress angiogenesis in ovo in colorectal cancer [102]. Altogether, the anti-angiogenic effect of *H. uninervis* further confirms its potential for developing therapeutics targeting TNBC tumor growth and metastasis.

Cancer is a multifaceted disease impacted by various factors and controlled by numerous signaling pathways. These pathways regulate the expression of diverse genes involved in the onset, progression, and metastasis of tumors. The STAT3 signaling pathway plays a crucial role in cell proliferation, migration, invasion, and angiogenesis [35]. The active form of STAT3 is constitutively expressed in many types of cancers, including breast cancer [103]. High levels of STAT3 contribute to tumor progression and metastasis in TNBC by regulating the expression of genes involved in cell cycle progression, apoptosis, EMT, and cell migration and invasion. A systemic examination of potential STAT3-regulated genes in aggressive TNBC revealed that the activation of STAT3 is associated with increases in cell cycle regulators (c-Fos, MEK5, c-Myc), apoptosis inhibitors (survivin, Mcl-1, Bcl-xL), and inducers of angiogenesis (VEGF, COX-2, MMP-2) [104]. Indeed, numerous plant-derived compounds have demonstrated their anticancer activities via the downregulation of the STAT3 signaling pathway and its regulated genes. For instance, the mistletoe (*Viscum album* L.) extract-induced inhibition of STAT3 promoted apoptosis, suppressed tumor growth, and inhibited lung metastasis in vivo via the downregulation of survivin, MMPs, and markers of EMT [105]. *Centipeda minima* extract also inhibited tumor progression and metastasis in MDA-MB-231 cells via the downregulation of multiple pathways, including STAT3 [106]. Moreover, extracts of *Origanum syriacum* and *Elaeagnus angustifolia* induced apoptosis in TNBC via the inhibition of STAT3 and activation of p53 [25,107]. In line with the aforementioned findings, HUE induced cell cycle arrest, promoted apoptosis, impeded cell adhesion, attenuated cell migration and invasion, and inhibited angiogenesis in MDA-MD-231 cells, potentially via the downregulation of the STAT3 signaling pathway and its targeted genes, including cell cycle regulators, apoptosis inhibitors, and inducers of angiogenesis (Figure 12).

While investigating the antioxidant potential and anticancer activity of HUE in vitro presented significant findings, this study has several limitations to be highlighted. Firstly, future work should focus on the fractionation of HUE crude extract to isolate and identify the bioactive metabolite(s) responsible for the observed effects. Additionally, further investigation of in vivo models is required to confirm the in vitro results and verify the effectiveness of HUE as an antioxidant and anticancer agent. Overall, our work, though important at the basic level, could be built upon in future research to elucidate the therapeutic potential of HUE. 

## 5. Conclusions

In conclusion, our findings indicate that the ethanolic extract of *H. uninervis* (HUE) exhibits potent antioxidant activity, as well as an anticancer effect on the malignant phenotype of TNBC, by targeting major hallmarks of cancer. HUE halted cell proliferation by inducing cell cycle arrest and activating intrinsic apoptosis. HUE also attenuated cancer metastasis by inhibiting cell migration, invasion, adhesion, and angiogenesis, potentially by targeting the constitutively active STAT3 signaling pathway. Moreover, HUE downregulated the levels of COX-2, a key player not only in angiogenesis but also in the inflammatory cascade. Given the established link between inflammation and cancer, evaluating the anti-inflammatory potential of HUE could pave the way for novel cancer treatments. It is worth mentioning that using the plant crude extract could offer more benefit over a single bioactive compound due to the synergistic effects present between various bioactives. Nonetheless, isolating and identifying the bioactive compound(s) in HUE constitute a significant ethnopharmacological approach and merit further investigation in future studies. Taken together, our study provides evidence for the potential anti-malignant effect of *H. uninervis* against TNBC, although further in vivo studies using animal models of breast cancer are still warranted.

## Figures and Tables

**Figure 1 antioxidants-13-00726-f001:**
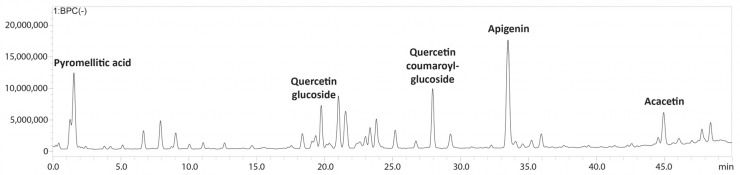
LC-MS/MS profile of *Halodule uninervis* ethanolic extract.

**Figure 2 antioxidants-13-00726-f002:**
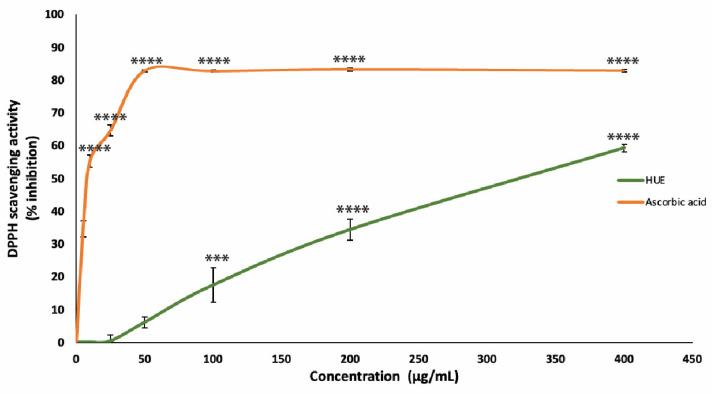
*Halodule uninervis* ethanolic extract has potent antioxidant capacity. The DPPH free-radical-scavenging assay was used to measure the antioxidant capacities of various concentrations of HUE (5, 10, 25, 50, 100, 200, and 400 μg/mL). Ascorbic acid was used as a reference. Values are represented as the mean ± SEM of three independent experiments. (*** *p* < 0.001 and **** *p* < 0.0001).

**Figure 3 antioxidants-13-00726-f003:**
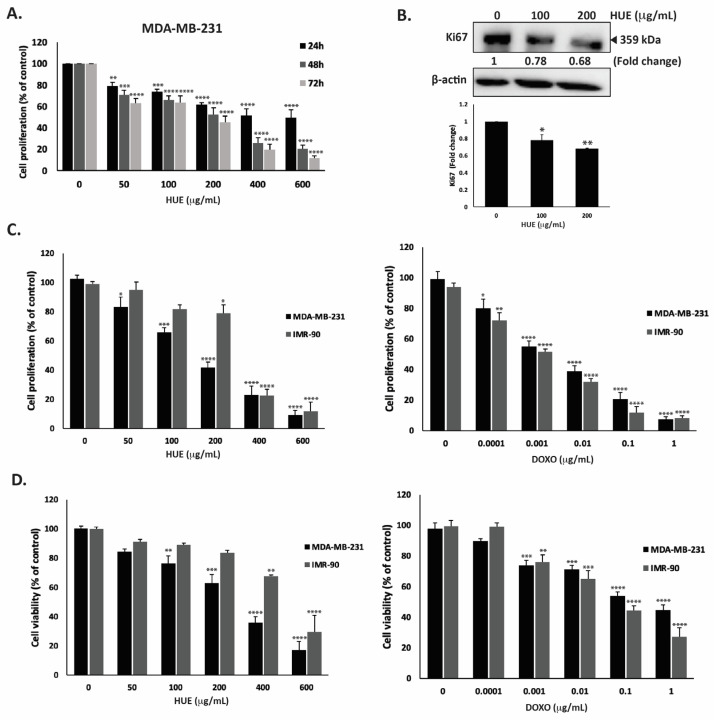
*Halodule uninervis* ethanolic extract exhibits an anticancer effect by inhibiting cellular proliferation and viability. (**A**) Time- and dose-dependent effects of HUE on the proliferation of MDA-MB-231 cells. Cell proliferation was examined using the metabolic-dye-based MTT assay. (**B**) MDA-MB-231 cells were treated for 24 h with and without the indicated concentrations of HUE. Cells were then lysed, and protein lysates were immunoblotted with a Ki67 antibody, using β-actin as a loading control. (**C**) Anti-proliferative effect on MDA-MB-231 and IMR-90 cells. Proliferation was measured after 72 h treatment with the vehicle control or the indicated concentrations of HUE/DOXO. (**D**) Cytotoxic effect on MDA-MB-231 and IMR-90 cells. Cytotoxicity was measured after 72 h of treatment with the vehicle control or the indicated concentrations of HUE/DOXO. Data represent the mean ± SEM of three independent experiments (*n* = 3) and are expressed as a percentage of the corresponding control cells. (* *p* < 0.05, ** *p* < 0.005, *** *p* < 0.001, **** *p* < 0.0001).

**Figure 4 antioxidants-13-00726-f004:**
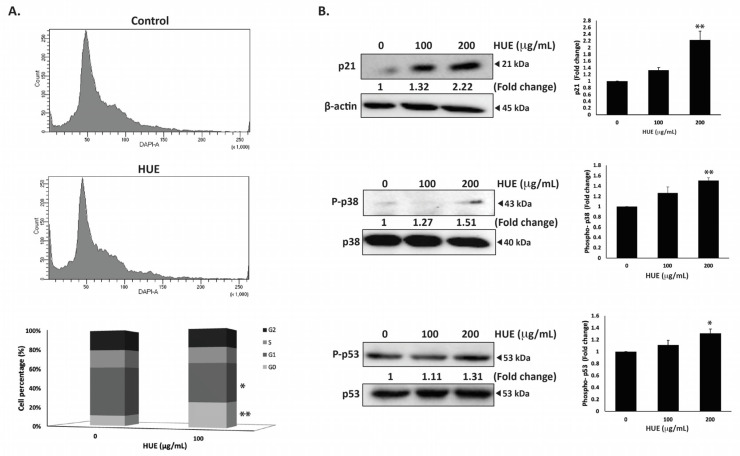
*Halodule uninervis* ethanolic extract induces G_0_/G_1_ cell cycle arrest in MDA-MB-231 cells. (**A**) MDA-MB-231 cells were incubated either with a vehicle-containing control or 100 μg/mL of HUE for 24 h. Cells were then collected, fixed, stained with 4′, 6-diamino-2-phenylindole (DAPI), and analyzed via flow cytometry. Data represent the mean ± SEM of three independent experiments (*n* = 3). (**B**) Cells were treated either with a vehicle-containing control or with HUE (100 or 200 μg/mL). Protein levels of phosphorylated p38, p21, and phosphorylated p53 were determined via Western blotting. β-actin was used as a loading control. Data represent the mean ± SEM of three independent experiments (*n* = 3). (* denotes *p* < 0.05, and ** *p* < 0.005).

**Figure 5 antioxidants-13-00726-f005:**
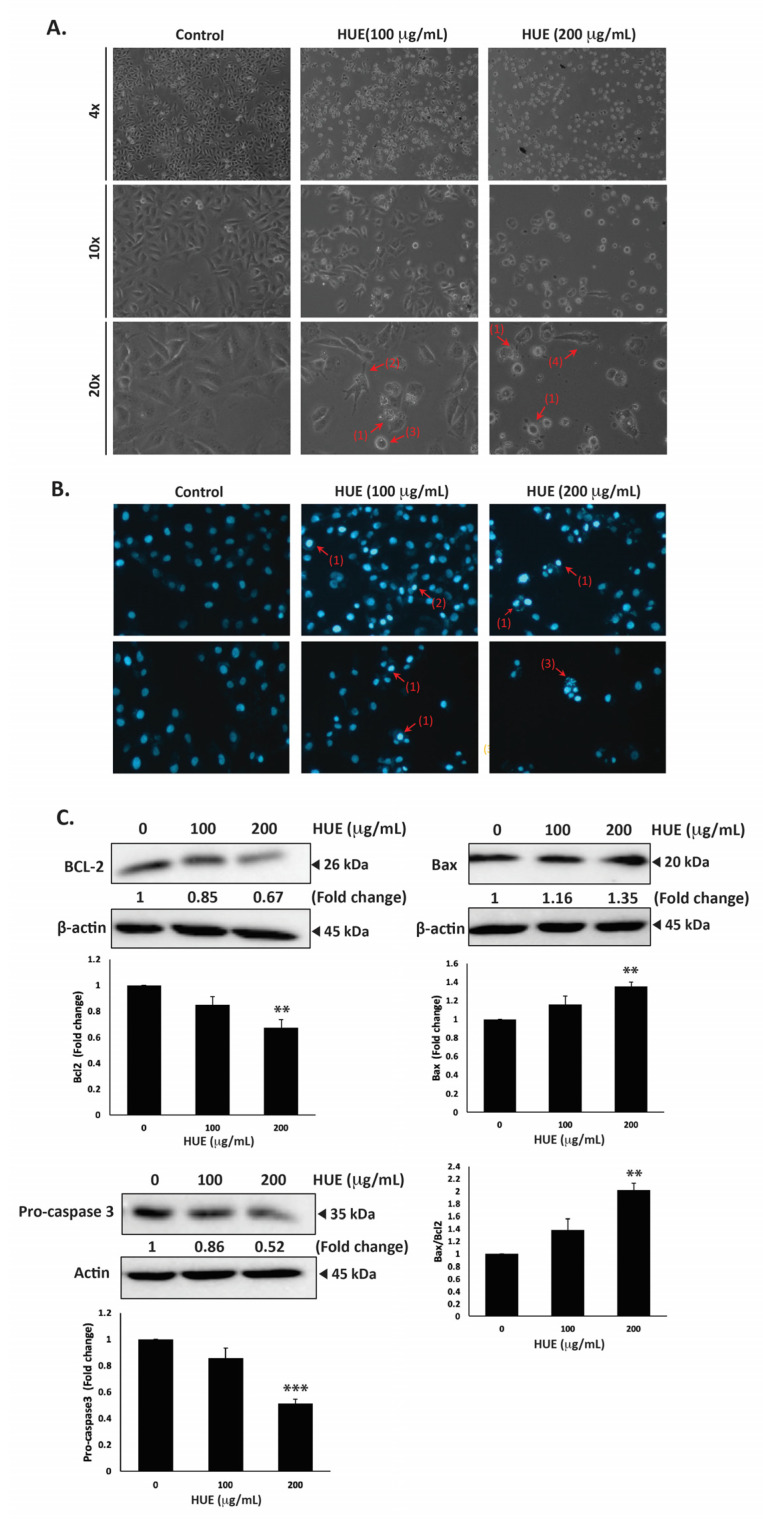
*Halodule uninervis* ethanolic extract induces apoptosis in MDA-MB-231 cells. (**A**) MDA-MB-231 cells were treated either with vehicle-containing control or HUE (100 or 200 μg/mL) for 24 h. Cellular morphological changes were detected via light optical microscopy. Arrows indicate (1) apoptotic bodies, (2) echinoid spikes, (3) membrane blebbing, and (4) cell shrinkage. (**B**) Cells were treated either with vehicle-containing control or HUE (100 or 200 μg/mL) for 24 h and stained with DAPI to visualize nuclear morphological changes via fluorescence microscopy. Pictures were taken at 10× magnification. Arrows show (1) chromatin lysis, (2) nuclear condensation, and (3) apoptotic bodies. (**C**) Cells were treated either with vehicle-containing control or HUE (100 or 200 μg/mL). Protein levels of procaspase 3, Bcl2, and Bax were determined via Western blotting. β-actin was used as a loading control. Data represent the mean ± SEM of three independent experiments (*n* = 3). (** denotes *p* < 0.005, and *** *p* < 0.001).

**Figure 6 antioxidants-13-00726-f006:**
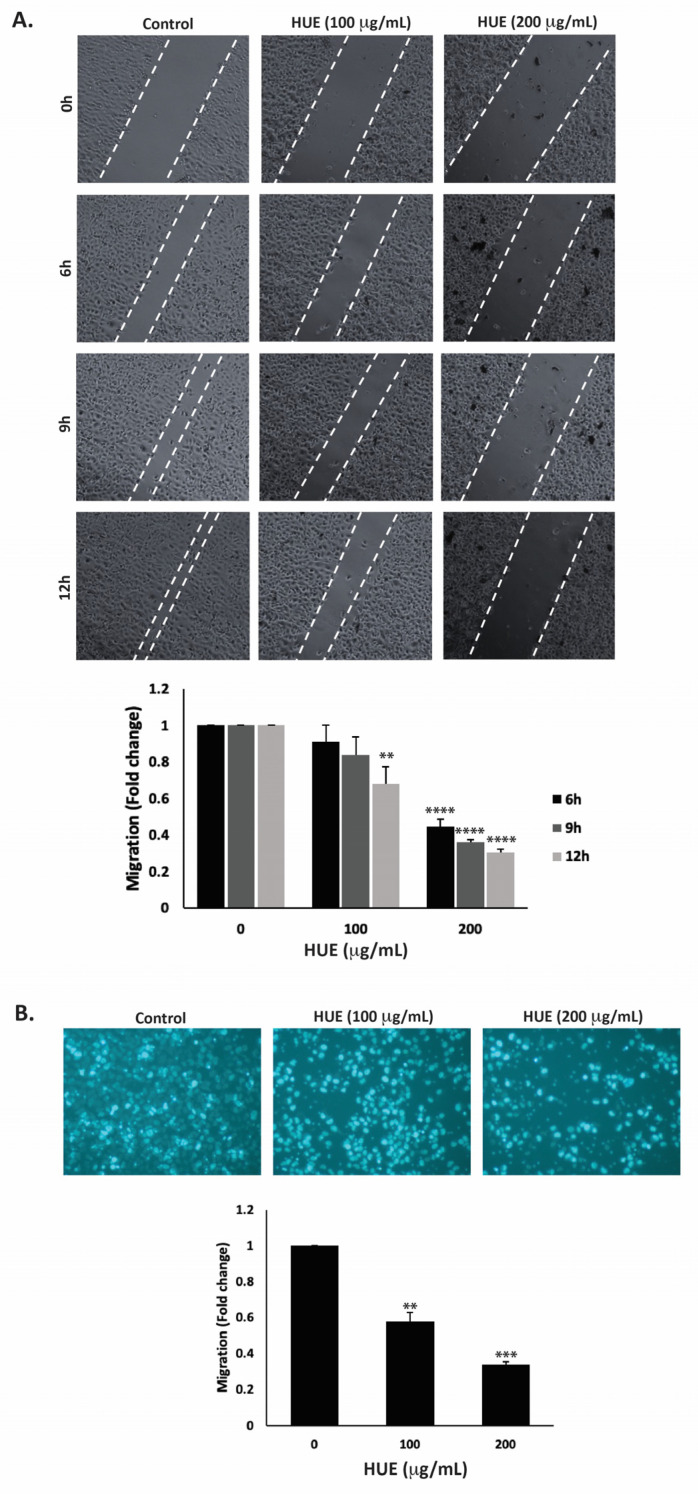
*Halodule uninervis* ethanolic extract inhibits the migration of MDA-MB-231 breast cancer cells. (**A**) Confluent cultures of MDA-MB-231 cells were wounded through scratching with a pipette tip. The cells were then incubated either with vehicle-containing control or HUE (100 or 200 μg/mL). Photomicrographs of the wound were taken using an inverted phase-contrast microscope at the indicated time points at 4× magnification. Values are plotted as the fold change in migration compared to vehicle-treated control cells. (**B**) MDA-MB-231 cells were treated overnight either with vehicle-containing control or HUE (100 or 200 μg/mL) in Boyden chamber trans-well inserts. Cells that migrated to the lower surface of the chamber were DAPI-stained, photographed at 4× magnification, counted, and analyzed. Data represent the mean ± SEM of three independent experiments (*n* = 3). (** *p* < 0.005, *** *p* < 0.001, **** *p* < 0.0001).

**Figure 7 antioxidants-13-00726-f007:**
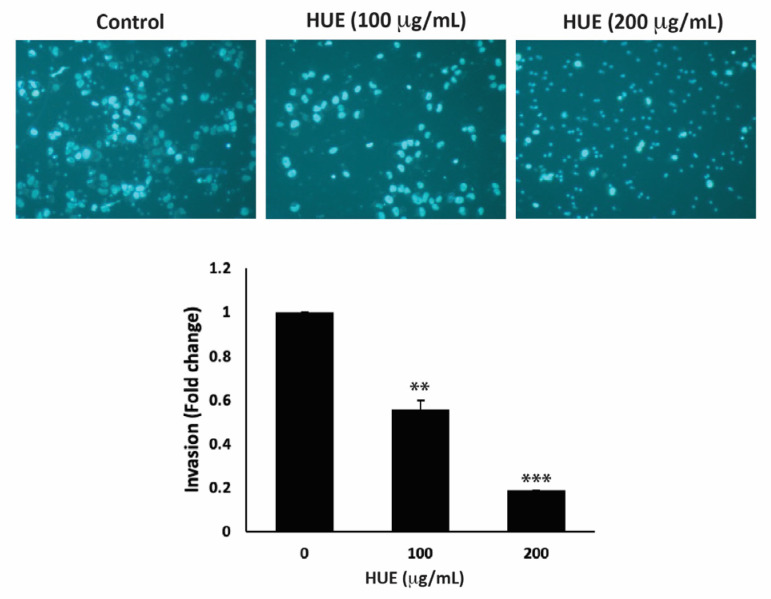
*Halodule uninervis* ethanolic extract reduces the invasive potential of MDA-MB-231 breast cancer cells. MDA-MB-231 cells were treated for 24 h either with vehicle-containing control or HUE (100 or 200 μg/mL) in Boyden chamber trans-well inserts pre-coated with Matrigel. Cells that invaded the Matrigel layer were DAPI-stained, imaged, counted, and analyzed. Pictures were taken at 4× magnification. Data represent the mean ± SEM of three independent experiments (*n* = 3). (** *p* < 0.005 and *** *p* < 0.001).

**Figure 8 antioxidants-13-00726-f008:**
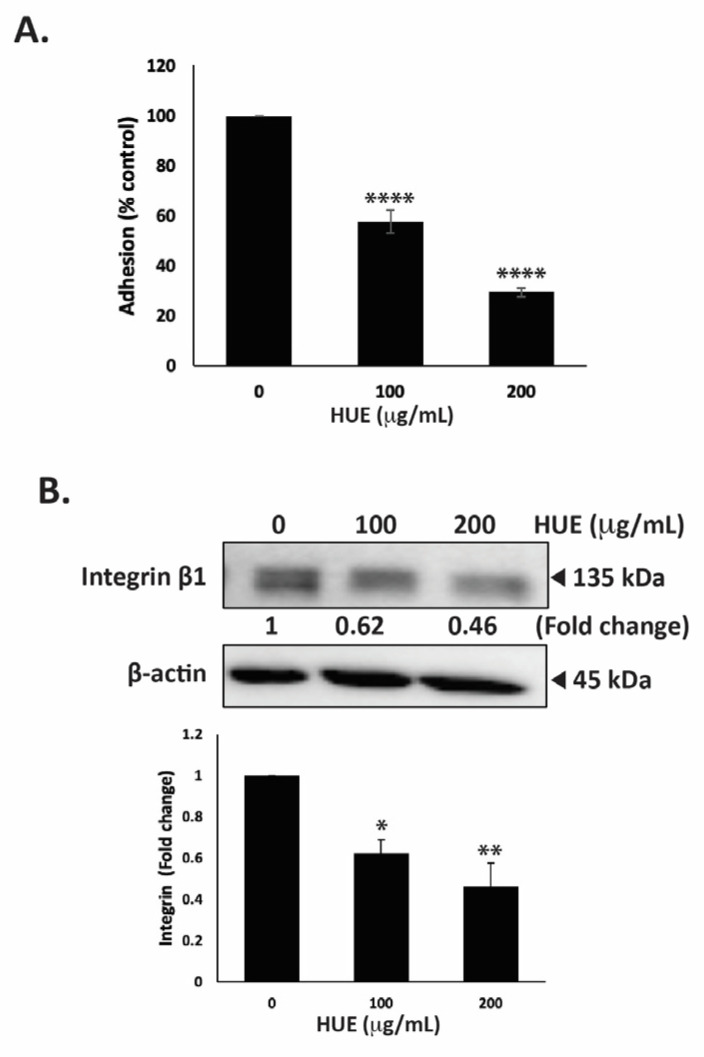
*Halodule uninervis* ethanolic extract inhibits the adhesion of MDA-MB-231 breast cancer cells to collagen and reduces integrin β1 protein levels. (**A**) MDA-MB-231 cells were incubated either with vehicle-containing control or HUE (100 or 200 μg/mL) for 24 h. Cells were then seeded onto collagen-coated wells and allowed to adhere for 3 h. Adhesion was assessed using the MTT assay and expressed as a percentage of the corresponding control cells. (**B**) MDA-MB-231 cells were incubated for 24 h either with vehicle-containing control or HUE (100 or 200 μg/mL). Whole-cell protein lysates were subjected to Western blotting analysis for integrin β1 expression levels, using β-actin as a loading control. Data represent the mean ± SEM of three independent experiments (*n* = 3). (* *p* < 0.05, ** *p* < 0.005, **** *p* < 0.0001).

**Figure 9 antioxidants-13-00726-f009:**
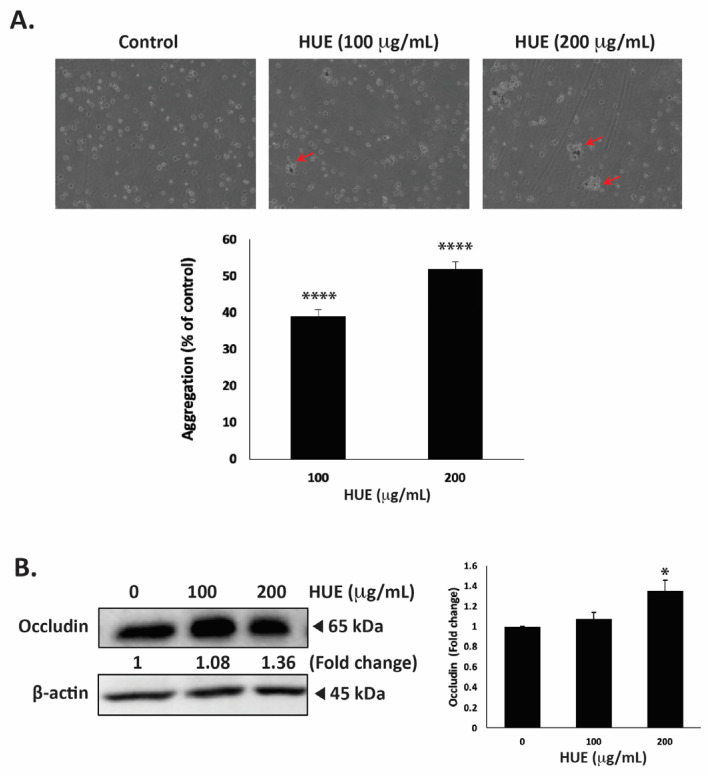
*Halodule uninervis* ethanolic extract increases cell–cell aggregation and reduces occludin levels in MDA-MB-231 breast cancer cells. (**A**) MDA-MB-231 cells were incubated either with vehicle-containing control or with HUE (100 or 200 μg/mL) and subjected to a cell aggregation assay. After 4 h, photomicrographs of cells were taken at 4× magnification, and the percentage of cell–cell aggregates was measured as indicated in the Materials and Methods section. Arrows point out to cell aggregates. (**B**) MDA-MB-231 cells were treated either with a vehicle-containing control or HUE (100 or 200 μg/mL) for 24 h. Western blotting was used to analyze the protein levels of occludin, using β-actin as a loading control. Data represent the mean ± SEM of three independent experiments (*n* = 3). (* *p* < 0.05, **** *p* < 0.0001).

**Figure 10 antioxidants-13-00726-f010:**
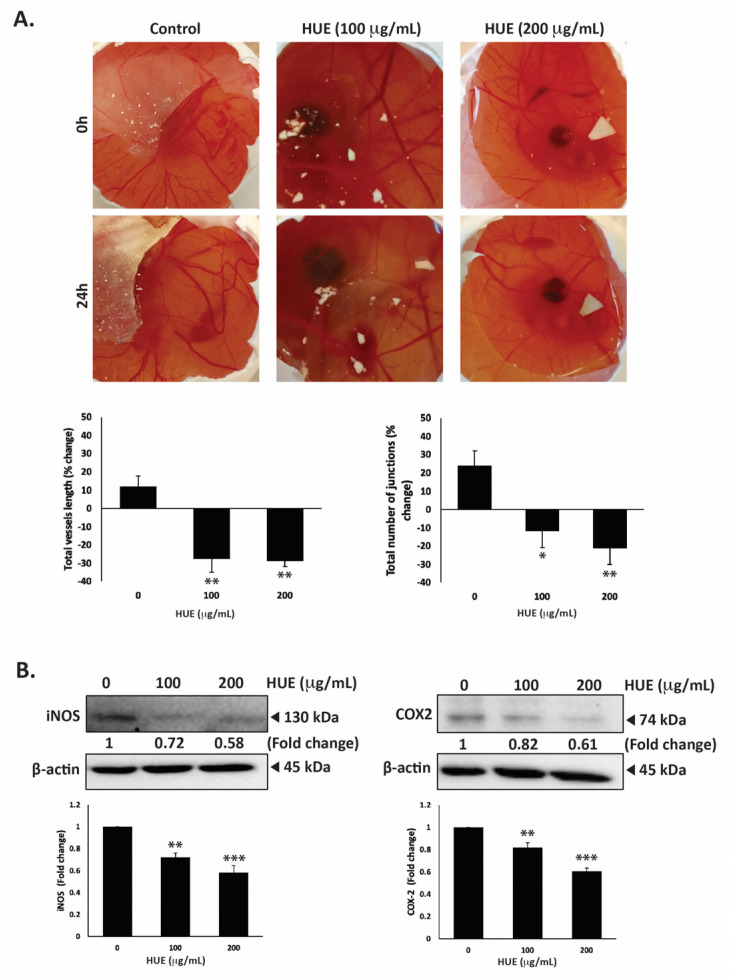
*Halodule uninervis* ethanolic extract inhibits angiogenesis in fertilized chicken eggs and reduces the levels of iNOS and COX-2 in MDA-MB-231 breast cancer cells. (**A**) HUE was applied to the chorioallantoic membrane (CAM) of fertilized chicken eggs for 24 h. Images were taken before and after treatment to score the angiogenic response. Total vessel length and the total number of junctions were measured using AngiTool 0.5a software and represented as the percentage change in vehicle-treated control. (**B**) MDA-MB-231 cells were treated for 24 h either with vehicle-containing control or HUE (100 or 200 μg/mL). The protein levels of iNOS and COX-2 were assessed via Western blotting, using β-actin as a loading control. Data represent the mean ± SEM of three independent experiments (*n* = 3). (* *p* < 0.05, ** *p* < 0.005, *** *p* < 0.001).

**Figure 11 antioxidants-13-00726-f011:**
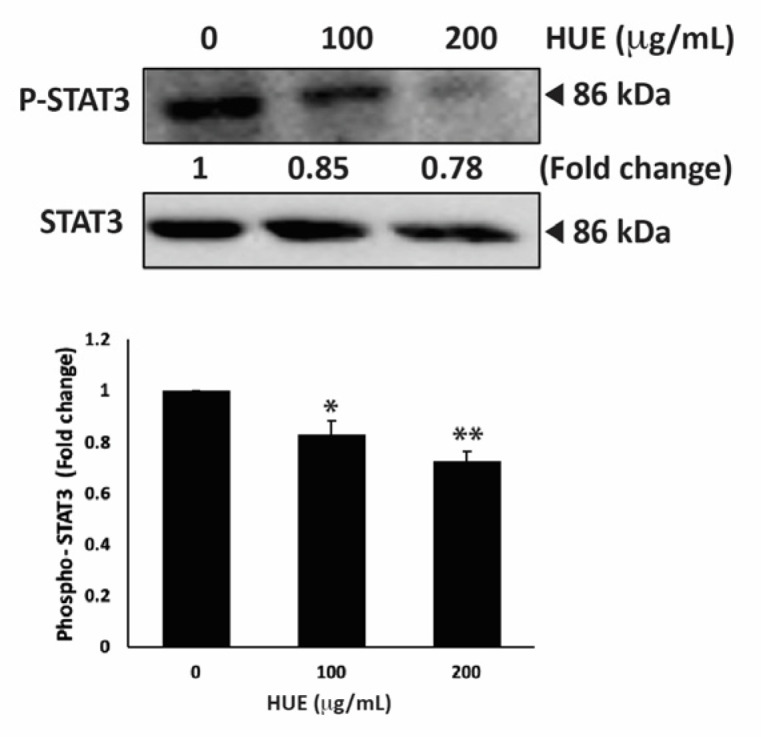
*Halodule uninervis* ethanolic extract inhibits the STAT3 signaling pathway. MDA-MB-231 cells were treated for 24 h either with vehicle-containing control or HUE (100 or 200 μg/mL). The protein levels of the phosphorylated STAT3 were assessed via Western blotting, using β-actin as a loading control. Data represent the mean ± SEM of three independent experiments (*n* = 3). (* *p* < 0.05 and ** *p* < 0.005).

**Figure 12 antioxidants-13-00726-f012:**
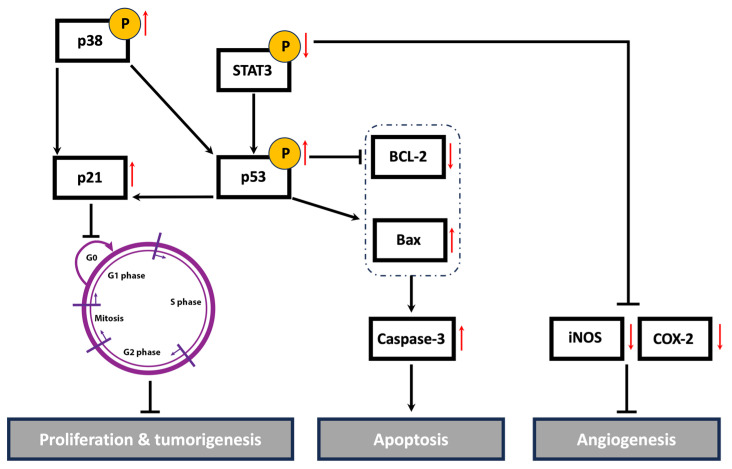
A schematic representation of the proposed anti-malignant mechanisms of *Halodule uninervis* ethanolic extract in MDA-MB-231 cells. HUE inhibited the hallmarks of cancer, possibly through targeting STAT3 and its associated proteins. Upward and downward red arrows indicate an increase and decrease in protein levels, respectively.

**Table 1 antioxidants-13-00726-t001:** Annotated compounds detected in *Halodule uninervis* ethanolic crude extract via LC-MS/MS.

N	Rt (min)	[M-H]^−^	MS/MS	Proposed Compound
1	1.57	253	139	Pyromellitic acid
2	6.56	153	108	Dihydroxybenzoic acid
3	8.98	137	108	Hydroxybenzoic acid
4	9.64	329	167	Vanillic acid glucoside
5	11.11	167	123	Vanillic acid
6	16.20	163	119	Coumaric acid
7	19.72	463	301	Quercetin glucoside
8	21.04	433	301	Quercetin pentoside
9	21.53	447	285	Kaempferol glucoside
10	22.92	431	269	Apigenin glucoside
11	23.74	461	284, 299	Diosmetin glucoside
12	23.83	417	285	Kaempferol pentoside
13	27.92	609	301, 463	Quercetin coumaroyl-glucoside
14	29.15	285	133	Kaempferol
15	29.53	593	285, 447	Kaempferol coumaroyl-glucoside
16	33.30	269	117	Apigenin
17	36.04	299	284	Diosmetin
18	44.78	283	268	Acacetin

**Table 2 antioxidants-13-00726-t002:** Phytochemical screening of *Halodule uninervis* ethanolic crude extract. (+) and (−) signs indicate the presence or absence of the specified secondary metabolite, respectively.

Metabolite	Results
Anthocyanins	−
Anthraquinones	−
Cardiac glycosides	+
Essential oils	+
Flavonoids	+
Phenols	+
Quinones	−
Resins	+
Saponins	+
Steroids	+
Tannins	+
Terpenoids	+

## Data Availability

The authors declare that the data supporting the findings of this study are available within the paper and its Appendix A files. Should any raw data files be needed in another format, they are available from the corresponding author upon reasonable request.

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
