# Peer review of "The Antioxidant Potential and Anticancer Activity of Halodule uninervis Ethanolic Extract against Triple-Negative Breast Cancer Cells"

_antioxidants, 2024, doi:10.3390/antiox13060726_

Round 1

Reviewer 1 Report

This is an interesting study in the context of the development of a potential new therapeutic approach for a difficult to treat cancer condition.

However, some problems could be identified:

-more details are necessary in experimental section – example: MTT assay (MTT concentration?). All text must be checked.

-authors referred that “HUE has high antioxidant capacity” – an “IC50 of 301.31 μg/mL” seems high and clearly ascorbic acid has higher antioxidant effect (in fact, what is the IC 50 for this reference compound?). This must be adequately analyzed and discussed in the paper.

-cell proliferation effects – authors must better justify why the studies focused on MDA-MB-2321 TNBC cell line. Do authors consider that these compounds are potent? It seems that IC50 values are high, again. In this context, I consider mandatory to include a positive control in this study and all these results analyzed and discussed in the paper.

-in addition, positive controls are relevant for the other cell-based studies.

-moreover, a non-tumoral cell line should be included in the study to evaluate the selectivity of these antiproliferative effects.

-formatting problems – example: “few drops of 2% FeCl3. Afterwards, 1 mL of concentrated sulfuric acid (H2SO4) was”. All text must be checked.

Author Response

Marseille, 01th June 2024,

Dear Reviewers, Dear Editor,

Thank you for your time and your insightful comments and feedback. We have taken all your suggestions into consideration and applied the necessary changes. We also went over the similarity report and paraphrased some words/sentences, which are tracked in the revised version with all other changes. We would like to highlight the fact that the similarity primarily stems from the materials and methods section and figures legends, which might be due to standard methods or commonly used language in the field. Nonetheless, the originality and integrity of our results and discussion remain intact. We hope the revised version of our manuscript will be found acceptable for publication in Antioxidants.

Regards

Dr Marc Maresca

Reviewer 1:

This is an interesting study in the context of the development of a potential new therapeutic approach for a difficult to treat cancer condition.

A: Dear Reviewer, thank you for your valuable input. We have made the necessary changes as requested.

However, some problems could be identified:

-more details are necessary in experimental section – example: MTT assay (MTT concentration?). All text must be checked.
A: Thank you for your suggestion. We have adjusted the experimental section as requested.

-authors referred that “HUE has high antioxidant capacity” – an “IC50 of 301.31 μg/mL” seems high and clearly ascorbic acid has higher antioxidant effect (in fact, what is the IC 50 for this reference compound?). This must be adequately analyzed and discussed in the paper.

A: Thank you for your question and comment. We have repeated the experiment using a lower range of concentrations to able to determine the IC50 of ascorbic acid. It was calculated to be 8.98 μg/mL. The DPPH assay result indicates that our crude extract has the potential for a significant antioxidant activity. We added a paragraph at the end of the discussion section about the limitations of our study, one of which was the fractionation of our crude extract. In future studies, we aim to fractionate the extract and identify the bioactive metabolites responsible for its observed antioxidant and anticancer activities.

-cell proliferation effects – authors must better justify why the studies focused on MDA- MB-2321 TNBC cell line. Do authors consider that these compounds are potent? It seems that IC50 values are high, again. In this context, I consider mandatory to include a positive control in this study and all these results analyzed and discussed in the paper.
A: Thank you for your comment and suggestion. We mentioned in the introduction the reason behind using the MDA-MB-231 cells. As it is a highly aggressive, with poor prognosis, and unresponsive to conventional cancer treatments, it is important to find alternative treatment approaches including those derived from plants and other natural resources. We have tested the antiproliferative effect of our crude extract on different cancer cell lines, revealing that it could possibly target the common hallmarks of cancer. Since we used a crude extract, containing many bioactive metabolites (as shown in our LC-MS/MS data), the IC50 was high as indicated by the reviewer. But based on the fact that MDA-MB cells are resistant to many anticancer drugs, having an activity of a plant extract is very interesting. We used Doxorubicin as positive control for anticancer effect and provide the data in the revised version.

-in addition, positive controls are relevant for the other cell-based studies.
A: Thank you for your suggestion. Unfortunately, it would take us up to 2-3 months to repeat all the cell-based studies with a positive control and we hope that the reviewer will understand we can’t perform such a long and expensive tests of redoing all tests during the revision.

-moreover, a non-tumoral cell line should be included in the study to evaluate the selectivity of these antiproliferative effects.
A: Thank you for your comment. In the revised version we tested the extract on normal human fibroblasts to check the selectivity and reported on that in the text.

-formatting problems – example: “few drops of 2% FeCl3. Afterwards, 1 mL of concentrated sulfuric acid (H2SO4) was”. All text must be checked.
A: Thank you for pointing such formatting problems out. We have gone over the manuscript and revised all formatting.

Regards

Reviewer 2 Report

Journal: Antioxidants

Manuscript ID: antioxidants-2989109
Type of manuscript: Article
Title: The antioxidant potential and anticancer activity of Halodule uninervis ethanolic extract against triple negative breast cancer cells 

Authors: Nadine Wehbe, Adnan Badran, Serine Baydoun, Ali Al-Sawalmih, Marc
Maresca, Elias Baydoun, Joelle Edward Mesmar

In the present research article, the antioxidant and anticancer potential of an ethanolic extract of the seagrass, Halodule uninervis (HUE), was investigated. It was found that HUE showed antioxidant property. In human triple-negative breast cancer (TNBC) MDA-MB-231 cells, HUE targeted cell proliferation, adhesion, migration, invasion, and angiogenesis to show its anti-proliferative and anti-metastatic activities, which could be mediated by down-regulation of the proto-oncogenic STAT3 signaling pathway. Thus, HUE may possess some promise for the discovery of anti-TNBC agents. This article reports detailed information about anticancer potentials of HUE, and it is thus recommended to be published as an article in Antioxidants after a few minor changes are considered by the authors, as shown below.

1.      Include LC-MS-MS profiles for analysis of the phytochemicals of HUE in Supporting Information (MS/MS spectra were missing in Figure 1).

2.      The authors need to test the major downstream proteins to verify the STAT3 signaling pathway (Figure 11), as proposed for HUE.  

Detailed comments for this manuscript are shown in the attached manuscript file. 

Author Response

Marseille, 01th June 2024,

Dear Reviewers, Dear Editor,

Thank you for your time and your insightful comments and feedback. We have taken all your suggestions into consideration and applied the necessary changes. We also went over the similarity report and paraphrased some words/sentences, which are tracked in the revised version with all other changes. We would like to highlight the fact that the similarity primarily stems from the materials and methods section and figures legends, which might be due to standard methods or commonly used language in the field. Nonetheless, the originality and integrity of our results and discussion remain intact. We hope the revised version of our manuscript will be found acceptable for publication in Antioxidants.

Regards

Dr Marc Maresca

Reviewer 1:

In the present research article, the antioxidant and anticancer potential of an ethanolic extract of the seagrass, Halodule uninervis(HUE), was investigated. It was found that HUE showed antioxidant property. In human triple-negative breast cancer (TNBC) MDA-MB-231 cells, HUE targeted cell proliferation, adhesion, migration, invasion, and angiogenesis to show its anti-proliferative and anti-metastatic activities, which could be mediated by down-regulation of the proto-oncogenic STAT3 signaling pathway. Thus, HUE may possess some promise for the discovery of anti-TNBC agents. This article reports detailed information about anticancer potentials of HUE, and it is thus recommended to be published as an article in Antioxidants after a few minor changes are considered by the authors, as shown below.

A: Dear Reviewer, thank you for your comments and sound feedback. We have made the requested changes in the manuscript and addressed the comments.

C1: Include LC-MS-MS profiles for analysis of the phytochemicals of HUE in Supporting Information (MS/MS spectra were missing in Figure 1).

A1: Thank you for your suggestion. Kindly note that MS/MS spectra are included in Figure 1. Table 1 also specifies the major phytochemicals present in the crude extract (name of the compound and retention time are specified).

C2: The authors need to test the major downstream proteins to verify the STAT3 signaling pathway (Figure 11), as proposed for HUE.

A2: Thank you for your comment. In this paper, we have demonstrated for the first time that HUE has an anti-breast cancer effect through STAT3 signaling as well as modulation of phosphorylation of associated proteins such as p53 or p21 and effect on the expression of downstream proteins such as Bax, Bcl2 or caspase as evidenced in Fig 5. We included the following schematic representation of the effect of HUE on STAT3 and associated proteins at the end of the revised manuscript.

C3: Results: HPLC-PDA-MS/MS: Detail the methods for identify these compounds, as presented in Table 1. Include these data MS/MS) in the supporting information.

A3: Thank you for your comment. We have expanded the HPLC-PDA-MS/MS methods section. In addition, we added the MS2 information for some of the identified compounds as a supplementary figure (Figure S1).

Regards

Reviewer 3 Report

In this work, authors have analyzed the composition in bioactive metabolites of the seagrass Halodule uninervis ethanolic extract and their antioxidant and anticancer effects against different tumour cell lines, in particular, the triple negative breast cancer MDA-MB-231 cells, They found that the anti-proliferative and anti-metastatic effects were associated with the downregulation of the proto-oncogenic STAT3 signaling pathway and conclude that H. uninervis could serve as a valuable source for developing novel drugs against this tumor.

The work appears mostly well done and full of tests and experiments to demonstrate the effects of this ethanolic extract.

However, the antioxidant and anticancer effects of extracts from this seegrass are already known, but not the effects on proliferation,  adhesion, migration, invasion, and angiogenesis of the triple negative breast cancer cells as well as the downregulation of the proto-oncogenic STAT3 signaling pathway.

 At first, the results of all these experiments demonstrate the antiproliferative action of the extract against tumor cells; however, it would be also useful to know if the action of this extract is also selective, without toxic effects or just very small toxicity towards normal cells, e.g. human fibroblasts or other non-tumorigenic cells.

This work also deals with the antioxidant action of Halodule uninervis ethanolic extract, so this implies that oxidation-reducing effects occur during the action of the compounds contained in it.

So, I wonder if it is appropriate to use the MTT reduction assay to evaluate cell viability with the risk of over-estimating or under-estimating the effect. As reported in the legend to Fig. 3, 80% ethanol was used as the vehicle control. But have the authors verified what happens by incubating the extract at the concentrations and times reported with MTT without cellular dehydrogenases that reduce MTT to formazan crystals?

There are many works that report the pitfall in the use of the MTT essay in cases like this. Wasn't it more appropriate to use an assay that does not involve redox interference, such as the crystal violet dye assay, which are based on the coloration of cellular biomass and not on redox reactions?

The cell cycle analysis in Fig.4A was conducted with 100 ug/mL of extract and for only 24 hr; however, the effects are not so marked and certainly less than those obtained with higher concentrations, e.g. 200 ug/mL as reported in the panels below and in the other figures.

Wouldn't it have been better to use conditions in which there would have been a more marked effect? Also because the analysis was conducted with only one parameter so the percentage of cells in the different phases of the cycle is not so precise. Please explain.

In Fig.3B, the concentrations 200 and 400 ug/mL in the WB are indicated, but the corresponding graph of the densitometric analysis below it shows different values. It should be an error, so fix it.

The panels in Figures 4 A-B and 5C seem small and readable only by zooming in, so I would suggest increasing their size.

The authors rightly write that it is worth mentioning that using the plant raw extract could offer more benefit over a single bioactive compound due to the synergistic effects present between various bioactives. Some of the compounds probably contained in the extract are known, such as apigenin, kaempferol, etc. therefore, one of these could perhaps be included in some experiments to compare its activity and thus corroborate the statement made.

The discussion appears informative but also very long, perhaps it is appropriate to reduce some parts to the most significant concepts and references.

Author Response

Marseille, 01th June 2024,

Dear Reviewers, Dear Editor,

Thank you for your time and your insightful comments and feedback. We have taken all your suggestions into consideration and applied the necessary changes. We also went over the similarity report and paraphrased some words/sentences, which are tracked in the revised version with all other changes. We would like to highlight the fact that the similarity primarily stems from the materials and methods section and figures legends, which might be due to standard methods or commonly used language in the field. Nonetheless, the originality and integrity of our results and discussion remain intact. We hope the revised version of our manuscript will be found acceptable for publication in Antioxidants.

Regards

Dr Marc Maresca

Reviewer 3:

C1: In this work, authors have analyzed the composition in bioactive metabolites of the seagrass Halodule uninervis ethanolic extract and their antioxidant and anticancer effects against different tumour cell lines, in particular, the triple negative breast cancer MDA-MB-231 cells, They found that the anti-proliferative and anti-metastatic effects were associated with the downregulation of the proto-oncogenic STAT3 signaling pathway and conclude that H. uninervis could serve as a valuable source for developing novel drugs against this tumor.

The work appears mostly well done and full of tests and experiments to demonstrate the effects of this ethanolic extract.

However, the antioxidant and anticancer effects of extracts from this seegrass are already known, but not the effects on proliferation, adhesion, migration, invasion, and angiogenesis of the triple negative breast cancer cells as well as the downregulation of the proto-oncogenic STAT3 signaling pathway.

 At first, the results of all these experiments demonstrate the antiproliferative action of the extract against tumor cells; however, it would be also useful to know if the action of this extract is also selective, without toxic effects or just very small toxicity towards normal cells, e.g. human fibroblasts or other non-tumorigenic cells.

This work also deals with the antioxidant action of Halodule uninervis ethanolic extract, so this implies that oxidation-reducing effects occur during the action of the compounds contained in it.

So, I wonder if it is appropriate to use the MTT reduction assay to evaluate cell viability with the risk of over-estimating or under-estimating the effect. As reported in the legend to Fig. 3, 80% ethanol was used as the vehicle control. But have the authors verified what happens by incubating the extract at the concentrations and times reported with MTT without cellular dehydrogenases that reduce MTT to formazan crystals?

There are many works that report the pitfall in the use of the MTT essay in cases like this. Wasn't it more appropriate to use an assay that does not involve redox interference, such as the crystal violet dye assay, which are based on the coloration of cellular biomass and not on redox reactions?

A1: Thank you for your comment. We have performed cell proliferation assay as well as cytotoxicity (cell viability) assay using our extract on MDA-MB-231 cells and on IMR-90 cells (normal human fibroblasts). We added the results to revised Figure 3 in our manuscript. We also performed the mentioned assays using Doxorubicin as a positive control. Results showed that, as DOXO, an known antiproliferative drug used in medicine, our extract was more active on dividing cells (antiproliferative assay) than on resting cells (cytotoxicity assay). This demonstrates that HUE inhibits cell division at concentration lower than the ones causing cell toxicity. In addition, HUE was found selective against MDA-MB-231 cells compared to IMR-90 cells while DOXO was nonselective and more toxic on normal cells. In addition, the 80% ethanol as vehicle is a typo mistake on our end. The extract stock solution was prepared in 80% ethanol. However, for the control the treatment was diluted, and the final concentration of ethanol was < 1% when applied to MDA-MB-231 cells. We have made this clear in Figure 3 legend. Finally, about the potential risk of using MTT assay to evaluate cell viability with the risk of over-estimating or under-estimating the effect, we performed other assay independent of dehydrogenases to quantify the number of cells in proliferation assay. We used ATP quantification (CellTiter-Glo) and protein quantification (BCA assay). In all case, at IC50, HUE reduced by around 50% the amount of ATP or proteins confirming a decrease in cell number caused by the extract and not an inhibition of cellular dehydrogenases.

C2: The cell cycle analysis in Fig.4A was conducted with 100 ug/mL of extract and for only 24 hr; however, the effects are not so marked and certainly less than those obtained with higher concentrations, e.g. 200 ug/mL as reported in the panels below and in the other figures.Wouldn't it have been better to use conditions in which there would have been a more marked effect? Also because the analysis was conducted with only one parameter so the percentage of cells in the different phases of the cycle is not so precise. Please explain.

A2: We thank you for your comment. As this is a sensitive experiment, we chose to use the 100 ug/mL concentration since the 200 ug/mL induced a higher apoptotic activity (shown in figure 5). Had we used the higher dose, more cells would be in the apoptotic state, and we wouldn’t have enough cells for the cell cycle analysis. On the other hand, we could have applied the treatment for less than 24 hours. However, this was the treatment time followed for all other experiments.

C3: In Fig.3B, the concentrations 200 and 400 ug/mL in the WB are indicated, but the corresponding graph of the densitometric analysis below it shows different values. It should be an error, so fix it.

A3: Thank you for noticing and pointing out this error. This has been fixed, and the figure was replaced.

C4: The panels in Figures 4 A-B and 5C seem small and readable only by zooming in, so I would suggest increasing their size.

A4: Thank you for pointing this out. The size has been adjusted, and the figures were replaced.

C5: The authors rightly write that it is worth mentioning that using the plant raw extract could offer more benefit over a single bioactive compound due to the synergistic effects present between various bioactives. Some of the compounds probably contained in the extract are known, such as apigenin, kaempferol, etc. therefore, one of these could perhaps be included in some experiments to compare its activity and thus corroborate the statement made.

A5: Thank you for your suggestion. In our future study, we aim to fractionate the crude extract and test the effects of the fractions on breast cancer cells. In parallel, the composition of the most active fraction will be investigated. As such, we will compare their effect to known compounds.

C6: The discussion appears informative but also very long, perhaps it is appropriate to reduce some parts to the most significant concepts and references.

A6: Thank you for your comment. We adjusted the discussion as suggested. We also moved some paragraphs to the introduction and conclusion.

Regards

Reviewer 4 Report

The manuscript entitled The antioxidant potential and anticancer activity of Halodule uninervis ethanolic extract against triple negative breast cancer cells brings important characteristics of HU which can be efficient in cancer treatment.

Observations

Introduction and Results

line 37 - replace conditions with disorders/diseases

line 46 - replace formidable with a proper synonym

line 57-58 - explain what is TNBC

line 60 - indicate the reference at the end of the sentence

Fig 1 - please indicate by labels the most important compounds visible in this figure

Discussions

line 587-606 - all this paragraph should be moved in Introduction

line 607-609 - indicate the figure here

As a general rule, everywhere where you are talking about your results, please insert, at the end of the sentence the figure, table or graphic you are discussing about.

line 621-623 - you already made this statement in the previous paragraph, please reformulate it.

line 683-686 - please indicate the reference here

At the end of Discussions please write about the limitation of this study and what you suggest would be useful to do in the next step of this research.

Would be better to make a separate chapter about conclusions to emphasize the results of your work and their importance for cancer therapy.

The manuscript entitled The antioxidant potential and anticancer activity of Halodule uninervis ethanolic extract against triple negative breast cancer cells brings important characteristics of HU which can be efficient in cancer treatment.

Observations

Introduction and Results

line 37 - replace conditions with disorders/diseases

line 46 - replace formidable with a proper synonym

line 57-58 - explain what is TNBC

line 60 - indicate the reference at the end of the sentence

Fig 1 - please indicate by labels the most important compounds visible in this figure

Discussions

line 587-606 - all this paragraph should be moved in Introduction

line 607-609 - indicate the figure here

As a general rule, everywhere where you are talking about your results, please insert, at the end of the sentence the figure, table or graphic you are discussing about.

line 621-623 - you already made this statement in the previous paragraph, please reformulate it.

line 683-686 - please indicate the reference here

At the end of Discussions please write about the limitation of this study and what you suggest would be useful to do in the next step of this research.

Would be better to make a separate chapter about conclusions to emphasize the results of your work and their importance for cancer therapy.

Author Response

Review 4:

The manuscript entitled The antioxidant potential and anticancer activity of Halodule uninervis ethanolic extract against triple negative breast cancer cells brings important characteristics of HU which can be efficient in cancer treatment.

Dear Reviewer, thank you for your comments. We have taken all your suggestions into consideration and made the required changes.

C1: Observations

Introduction and Results

  1. line 37 - replace conditions with disorders/diseases

A: Thank you for your suggestion. We replaced it with “disorders”.

  1. line 46 - replace formidable with a proper synonym

A: Thank you for your suggestion. We replaced it with “challenging”.

  1. line 57-58 - explain what is TNBC

A: Thank you for your comment. Clarification on the explanation of TNBC has been added lines 65-69. Hoping this is clear now and enough.

  1. line 60 - indicate the reference at the end of the sentence

A: Thank you for pointing out the missing reference. The reference was added.

  1. Fig 1 - please indicate by labels the most important compounds visible in this figure

A: Thank you for your suggestion. We have added some of the compounds on the MS/MS spectrum (Figure 1).

Discussions

  1. line 587-606 - all this paragraph should be moved in Introduction

A: Thank you for your suggestion. The paragraph has been moved to the introduction lines 42-56.

  1. line 607-609 - indicate the figure here. As a general rule, everywhere where you are talking about your results, please insert, at the end of the sentence the figure, table or graphic you are discussing about.

A: Thank you for your suggestion. We have added the corresponding figures to the discussed results.

  1. line 621-623 - you already made this statement in the previous paragraph, please reformulate it.

A: Thank you for pointing this out. We made the necessary changes to avoid repetitions.

  1. line 683-686 - please indicate the reference here.

A: Thank you for pointing this out. This has been adjusted.

  1. At the end of Discussions please write about the limitation of this study and what you suggest would be useful to do in the next step of this research.

A: We thank you for this suggestion. We have added a new paragraph to the end of the discussion talking about the limitations of our study.

  1. Would be better to make a separate chapter about conclusions to emphasize the results of your work and their importance for cancer therapy.

A: Thank you for your suggestion. We moved the last paragraph in the discussion to a new section – Conclusion.

Round 2

Reviewer 1 Report

No major comments. Authors addressed the majority of issues indicated.

No detail comments. Authors addressed the majority of issues indicated.

Author Response

Dear Reviewer,

As requested, we now provide the data on ATP and proteins quantification directly in the main text of the revised manuscript on page 10.

"

Using MTT assay, the IC50 was 525.34, 173.37, and 146.74 µg/mL at 24, 48, and 72 h. Based on the IC50 values, 100 and 200 μg/mL HUE were used in further experiments. In order to confirm that the observed decrease in MTT metabolisation corresponds to a real decrease in cell number and not to an enzymatic inhibition of the cellular dehydrogenases by HUE, other assays were performed. Quantification of ATP (by CellTiter-Glo) and proteins (by BCA assay) contents in MDA-MB-231 cells exposed for 72 h to HUE at 200 µg/mL showed 59.92 +/- 5.62 and 52.63 +/- 7.98 % decrease compared to 58.22 +/- 6.61 % decrease for MTT signal, demonstrating that HUE indeed inhibits cell division rather than inhibiting cellular dehydrogenases."

Regards

Reviewer 3 Report

The authors responded quite comprehensively to the reviewer's requests. The only thing that remains unclear is why they did not report at least as supplementary files the ATP quantification (CellTiter-Glo) and protein quantification (BCA assay) data mentioned in answer A1.

The authors responded quite comprehensively to the reviewer's requests. The only thing that remains unclear is why they did not report at least as supplementary files the ATP quantification (CellTiter-Glo) and protein quantification (BCA assay) data mentioned in answer A1.

Author Response

Dear Reviewer,

We do not see your comments in the second round of reviewing.

Regards